# Recruitment Strategies That Take a Chance

**Gregory Kehne**
Harvard University

**Ariel D. Procaccia**
Harvard University

**Jingyan Wang**
Georgia Institute of Technology

## Abstract

In academic recruitment settings, including faculty hiring and PhD admissions, committees aim to maximize the overall quality of recruited candidates, but there is uncertainty about whether a candidate would accept an offer if given one. Previous work has considered algorithms that make offers sequentially and are subject to a hard budget constraint. We argue that these modeling choices may be inconsistent with the practice of academic recruitment. Instead, we restrict ourselves to a single batch of offers, and we treat the target number of positions as a soft constraint, so we risk overshooting or undershooting the target. Specifically, our objective is to select a subset of candidates that maximizes the overall expected value associated with candidates who accept, minus an expected penalty for deviating from the target. We first analyze the guarantees provided by natural greedy heuristics, showing their desirable properties despite the simplicity. Depending on the structure of the penalty function, we further develop algorithms that provide fully polynomial-time approximation schemes and constant-factor approximations to this objective. Empirical evaluation of our algorithms corroborates these theoretical results.

## 1 Introduction

Anyone who has served on a faculty hiring committee or a PhD admissions committee knows that a successful outcome requires resolving the tension between two competing goals. On the one hand, some candidates are (perceived to be) better qualified than others, and the aim is to recruit the best candidates. On the other hand, there are a given number of positions to be filled, and while there is typically some flexibility, there is a real cost to recruiting too many or too few people. The tension arises in part because the stronger a candidate is, the more likely they are to receive multiple attractive offers and the less likely they are to accept any particular offer. In order to manage uncertainty, a good strategy may involve a mix of offers to stellar candidates and "safer" candidates.

To formalize this problem, we assume that a recruiting entity (academic or otherwise) has access to two numbers for each candidate $i$: their value $x_i$ and their probability $p_i$ of accepting an offer. We acknowledge that in current practice, these numbers are not always explicitly estimated. However, committees typically rank or assign numerical scores to candidates based on their strength or fit, and savvy committees roughly estimate recruitment chances by classifying candidates as, say, "high yield," "low yield" or "extremely low yield", for example, by past experience or assistive computational tools [13, 1]. Therefore, we believe that the gap between current practice and explicit value and probability estimates is not large.

Our approach builds on the work of Purohit et al. [12], who cast hiring under uncertainty as a stochastic optimization problem. In their basic model, there are $n$ candidates (each associated with a value and probability), $k$ positions, and $t$ time steps. In each time step, the algorithm (i.e., recruitment strategy) may make an offer to a single candidate and receive a response; that is, at most $t$ sequential offers can be made, and the budget of $k$ cannot be exceeded. The goal is to maximize the expected value of candidates who accept offers. Purohit et al. [12] also consider the setting where the algorithm may make parallel offers in each round. For both problems, they develop polynomial-time, constant-factor approximation algorithms (with approximations ratios of 2 and 8, respectively).

36th Conference on Neural Information Processing Systems (NeurIPS 2022).

This problem formulation captures key aspects of recruitment, but, in our view, it does have two shortcomings. First, in a sense it is overcomplicated, as computational challenges stem from the assumption that offers are made sequentially. But, in our experience of faculty hiring and PhD admissions in several universities, offers are typically made in one batch. Indeed, delayed offers (in the case of faculty hiring) and waitlists (in the case of PhD admissions) are usually avoided as they negatively impact yield.

The second, and more crucial, shortcoming is that Purohit et al. [12] consider the constraint of hiring $k$ candidates as firm. Again, this is inconsistent with our experience: offers are made so that the expected yield roughly matches a desired target, but some faculty hiring or PhD admission cycles are "too successful," in the sense that the number of candidates who accept their offers is much larger than expected. This has a real cost: in the case of PhD students, it creates difficulties in finding funding and advisors, and in the case of faculty hiring, it may precipitate a shortage of resources with long-term impacts on future hiring and even tenure. For example, in one of our institutions, a faculty hiring cycle with yield that was much higher than expected led to the cancellation of the subsequent year's search.

Let us, therefore, reformulate the problem of hiring under uncertainty in a way that avoids both issues. We assume that offers are made in a single batch, and the target number of positions $k$ is a soft constraint. Specifically, a penalty is incurred for deviating from the target number of positions; we consider several different options for this penalty function. The optimization problem is this:

> *Select a subset $S$ of candidates that maximizes overall expected reward $\sum_{i \in S} p_i \cdot x_i$, minus expected penalty for deviating from the target number of positions.*

Enumerating all possible subsets $S$ may be practicable for small instances, for example in the case of faculty hiring in small departments. However, a brute-force approach will not work for this purpose in larger departments, or at the scale of PhD admissions even in smaller programs, which motivates our search for good algorithms.

**Our results.** We first consider a simplified case where the goal is to solely minimize the penalty term of our objective (irrespective of the rewards), and show that the greedy algorithm that selects candidates in decreasing order of their probabilities is optimal (Section 3.1).

The full objective is considerably more complex, and we analyze it under two natural penalty functions. When the penalty function is the squared error from the target, we show that the optimization problem is weakly NP-hard, and provide a fully polynomial-time approximation scheme (FPTAS). When the penalty is linear in the extent to which the target is exceeded (that is, a linear penalty is incurred by overshooting, but not by undershooting), we show that two greedy heuristics — picking in the decreasing order of the value $x_i$ and the expected value $p_i x_i$ — provide approximations to the optimal solution that are polynomial in the minimum probability $p_{\min}$. We then present a constant-factor approximation algorithm that runs in polynomial time for fixed $p_{\min}$ and candidate value relative to overshooting penalty, thereby improving upon the greedy heuristics.

Finally, we carry out experiments on synthetically generated data (Section 4), focusing on the linear penalty incurred by overshooting. We observe that the two greedy heuristics perform reasonably well, especially if the values and probabilities are positively correlated. At the same time, compared to the greedy heuristics, our constant-factor approximation algorithm better adapts to specific instances, especially when this correlation is negative. These numerical experiments corroborate our theoretical results that the greedy heuristics provide reasonable guarantees, justifying their use in practice for both simplicity and good performance. However, in many regimes the additional flexibility of the constant-factor approximation algorithm is likely worth its complexity overhead.

**Related work.** In practice, the challenge of uncertainty in admissions is mitigated by practices such as admitting students in multiple rounds, using a waitlist, and using a rolling process [9]. There are also assistive computational tools that predict student yield rate with machine learning [13, 1]. There are a few theoretical formulations that model and address the uncertainty in such problems. As previously mentioned, Purohit et al. [12] consider an online setting where in each time step if a candidate is given an offer then their decision is revealed immediately, and analyze the optimal ordering to give offers to the candidates subject to a hard constraint on the total number of acceptances. Ganguly et al. [7] consider a setting with multiple rounds where the yield rate in each round is either

$q_L$ or $q_H$ (with $q_L < q_H$). To model the negative correlation between the candidate quality and the probability of acceptance, they assume that the probability of the yield rate being $q_H$ is linear in the number of students given offers. They subsequently derive a decision tree that computes the number of offers to make in each round. For single batch selection, Zhang and Pippins [16] analyze the optimal number of applicants to admit using techniques from yield management, under the assumption that each applicant has identical value and probability. A distinct line of work casts the admissions problem as a decentralized matching market [4, 5], where the uncertainty in acceptance is modeled by the students' stochastic preferences over multiple schools. An objective combining the utility and the accepted size is considered, but the penalty is in terms of the expected size, and does not consider variance. Another line of work analyzes various metrics for the secretary problem under uncertain offer acceptance [14, 15, 11], where candidates appear sequentially in a uniformly random ordering in a fleeting nature. Here the reward is defined based on the ranking of the candidate accepting the offer relative to other candidates in the sequence, and the goal is to decide whether to make an offer to the candidate appearing in each time step.

Our proposed formulation is closely related to the knapsack problem and its many variants. In the stochastic knapsack problem, each item has a deterministic value and an independent stochastic size; the actual size of an item is revealed only after it is selected [8, 6, 2]. For the one-shot version of this problem, the aim is to choose a subset maximizing the expected value of the realized items, such that the probability that these realizations violate the knapsack constraint is below some threshold. In contrast, our "items" have equal size, but we pay a penalty which is some function of our realized distance from our knapsack target size, exchanging the constraint for a mixed-sign objective. In this spirit, the one-sided loss functions we consider are similar to objectives which arise in the penalty method for solving constrained optimization problems [10], though our setting is stochastic and we do not introduce the penalty term in service of ultimately satisfying a hard constraint.

## 2   Problem Formulation

Taking a knapsack perspective, consider some $n$ items (corresponding to candidates) with associated values $x_1, \ldots, x_n \in \mathbb{R}$. If we select an item $i \in [n]$, there is a probability $p_i \in [0, 1]$ that we receive this item (the candidate accepts the offer). We write these values and probabilities as vectors $x \in \mathbb{R}^n$ and $p_i \in [0, 1]^n$. Let $Z_i \in \{0, 1\}$ be the indicator variable that we receive item $i$ if it is selected, so that $Z_i \sim \mathrm{Ber}(p_i)$. We assume the events that we receive each individual item are independent. Let $S_Z \subseteq S$ denote the random realization of chosen items; that is, $S_Z := \{i \in S : Z_i = 1\}$. Our goal is to select a subset $S \subseteq [n]$. First, we consider the reward for a subset $S$ as the expected total value obtained:

$$R(S) := \mathbb{E}\left[\sum_{i \in S} Z_i x_i\right] = \sum_{i \in S} p_i x_i.$$

At the same time, let $M \in \mathbb{N}_+$ denote a target size that we want the realized set $S_Z$ to achieve. We want to control the expected deviation of the realized size of $S_Z$, which is $|S_Z| = \sum_{i \in S} Z_i$, from the target size $M$. We consider this penalty as

$$V(S) := \mathbb{E}\left[\rho\left(|S_Z|,\ M\right)\right],$$

where $\rho \colon \mathbb{N} \times \mathbb{N}_+ \to \mathbb{R}_{\geq 0}$ is a loss function, to be specified later. Combining the two parts, we define the overall objective as

$$U(S) := R(S) - \lambda \cdot V(S), \tag{1}$$

where $\lambda \in \mathbb{R}_+$ is a hyperparameter that governs the importance of the penalty relative to the reward. Our goal is to find the subset that maximizes the overall expected utility:

$$S^* \in \arg\max_{S \subseteq [n]} U(S).$$

We denote a problem instance by $\mathcal{I} := (x, p, M, \lambda)$, and the solution $S^*$ is thus a function of the problem instance and the loss function $\rho$. It is worth noting that since the overall utility $U$ is a mixed-sign objective, the optimal value of $U(S^*)$ may be negative depending on the instance and choice of $\rho$.

We consider a range of choices for the loss function $\rho$. Given a target size $M$, it is natural for $\rho$ to be a convex function minimized at $M$, which penalizes any deviation from $M$, or alternatively a

monotone convex function that is nonzero above $M$, which can be seen as penalizing violation of a budget constraint. We focus in particular on one- and two-sided linear and quadratic losses, which are formally introduced in Section 3.2 below.

# 3 Theoretical Results

To begin we note that if we only consider the reward term and set the penalty term to be $V(S) := 0$, then the solution is to trivially select all items. In what follows, we first discuss the other extremal case, taking $R(S) := 0$ and considering the penalty $V$ in isolation. These may be viewed as the extreme cases when $\lambda = 0$ and $\lambda \to \infty$. We will then turn to the general objective and consider both terms jointly.

## 3.1 Warm-Up: Penalty Only

To gain intuition for this problem, we start with the simplified case in which our goal is only to minimize the penalty term. Note that in this case our objective is strictly nonpositive.

---

**Algorithm 1** PGREEDY

---

**Require:** $p \in [0, 1]^n$
1: $S \leftarrow \emptyset$.
2: Sort $\{p_i\}_{i \in [n]}$ in decreasing order and re-index the items such that $p_1 \geq \ldots \geq p_n$
3: **for** $i = 1, 2, \ldots, n$ **do**
4:     **if** $U(S \cup \{i\}) \geq U(S)$ **then**
5:         $S \leftarrow S \cup \{i\}$
6:     **else**
7:         **break**
8: **return** $S$

---

We consider PGREEDY, the greedy algorithm with respect to $p_i$ (Algorithm 1). In words, PGREEDY selects items in their decreasing order of probabilities, with ties broken arbitrarily if there are multiple items with the same probability.[1] The algorithm keeps selecting the next item defined by this order, and terminates when adding the next item would decrease the objective. This greedy algorithm is computationally efficient, since the stopping criterion in Line 4 can be checked in polynomial time given access to $\rho$ (see Lemma 5 in Appendix B.8 for details). Surprisingly, PGREEDY is optimal for minimizing $V(S)$ in isolation.

**Proposition 1.** *Let $M \in \mathbb{N}_+$ be any target size. If the loss function $\rho(\cdot, M)$ is convex, then* PGREEDY *(Algorithm 1) yields an optimal solution to minimizing the penalty $\min_{S \subseteq [n]} V(S)$.*

The proof of this proposition is provided in Appendix B.3. This result is not obvious, as one might expect that as the sum of probabilities of all selected items so far approaches the target size, it may be better to select an item with lower probability than an item with higher probability to "fill the gap." This is not true. Intuitively, it is because the realization of each acceptance $Z_i$ is binary, so the outcome of adding another item $i$ into the selection is either we add this item (with probability $p_i$) or not (with probability $1 - p_i$). If adding this item gives lower penalty, then we desire to add the item with the highest probability possible.

## 3.2 The General Objective

We now turn to the general objective. At the outset it bears noting that $U(S)$ is submodular in $S$ whenever $\rho(\cdot, M)$ is convex (see Lemma 1 in Appendix B.1), as is the case for the loss functions we consider. Unfortunately the existing body of work on (non-monotone) submodular maximization cannot be leveraged to obtain a general-purpose approximation to $U(S)$, since $U$ is mixed-sign and may be negative even at optimality, and applying an affine transformation in order to engineer nonnegativity will generally destroy any approximation guarantees.

We focus on a few natural choices of the loss function. First, we consider $\rho$ given by linear and quadratic losses, which we denote by $L_1$ and $L_2$ respectively. These yield penalty terms $V(S)$ which

---

[1]In practice it is natural to break ties in favor of items of higher value, though this does not affect our results.

are equal to the mean average error (MAE) and mean squared error (MSE) for the realized size of the subset $S_Z$. We also consider the corresponding one-sided losses, defined by

$$L_1^+(|S_Z|, M) := \begin{cases} |S_Z| - M & \text{if } |S_Z| \geq M \\ 0 & \text{otherwise,} \end{cases} \quad \text{and} \quad L_2^+(|S_Z|, M) := L_1^+(|S_Z|, M)^2.$$

All of these losses considered penalize the case where the realized size is greater than the target size $M$. In applications such as admissions and hiring, there is a limited, pre-specified amount of resources allocated to the newly admitted or hired people. Hence, having more people than the target size is not desired. At the same time, the two-sided losses give explicit preference that the realized size should also not be smaller than the target size. This explicit penalty for undershooting could represent a hit to morale (an unsuccessful recruitment cycle really is demoralizing) or insufficient staffing for required tasks, such as teaching certain courses. The one-sided loss functions may still to some extent capture these considerations, as there is an implicit opportunity cost described by the reward term when fewer candidates accept.

### 3.2.1 An FPTAS for $L_2$ Loss

Given that we understand our problem in both extremal cases (when only considering the reward term or the penalty term), one might hope that some interpolation between them could solve the general case. However, the general case is more complicated. Recall that for the penalty-only objective, PGREEDY in Algorithm 1 attains the optimal selection, by adding items in decreasing order of $p_i$, and terminates once the next item strictly decreases the objective. But PGREEDY is clearly ill-suited to the general objective, since it does not take values into consideration. We now present two more natural greedy heuristics analogous to PGREEDY, and show that they are provably not optimal for the general objective. Specifically, we consider:

- XGREEDY: adds items in decreasing order of their value $x_i$, and terminates once the next item in this order strictly decreases the objective.

- XPGREEDY: adds items in decreasing order of their expected reward $x_i p_i$, and terminates once the next item in this order strictly decreases the objective.

Despite these heuristics appearing intuitive, they perform in a certain sense arbitrarily poorly even for the (squared) $L_2$ loss, as formalized by the following result.

**Proposition 2.** *Consider the two-sided loss $\rho = L_2$ and any $\lambda > 0$. Then for PGREEDY, XGREEDY and XPGREEDY, there exists an instance such that the algorithm selects $S \subseteq [n]$ for which $U(S) \leq 0$, while $U(S^*) > 0$.*

The proof of this proposition is provided in Appendix B.4, and we now provide an informal description of the instances constructed. Since PGREEDY does not take into account the item values $x_i$ at all, it is natural to expect that PGREEDY is not suitable for the general objective. Specifically, we consider two items where one item has probability 1 and value 0, and the other item has a "good" probability less than 1 and a "good" value. Then PGREEDY selects the first item, whereas selecting the second item yields a positive objective value. For XGREEDY and XPGREEDY, we consider two items that have almost the same expected reward. We let item 1 has a probability of 1. We let item 2 have a slightly greater expected reward for tie-breaking, and let item 2 have a smaller probability. In this case, XGREEDY and XPGREEDY start by picking item 2, which introduces nontrivial variance. When $\lambda$ becomes large, this variance drives the overall objective negative. On the other hand, picking item 1 yields a strictly positive objective.

Despite the failure of heuristic approaches, when the chosen loss function is $\rho = L_2$ our problem can in fact be approximated up to negligible additive error. For this loss, our full objective (1) may be written as

$$U(S) = \sum_{i \in S} p_i x_i - \lambda \cdot \mathbb{E}\left(\sum_{i \in S} Z_i - M\right)^2. \tag{2}$$

Letting $b_i \in \{0,1\}$ be the binary decision variable for whether $i \in S$, that is, $b_i := \mathbb{1}\{i \in S\}$, the optimization problem then becomes

$$\underset{S \in [n]}{\arg\max} \, U(S) = \underset{b \in \{0,1\}^n}{\arg\max} \, \sum_{i \in [n]} b_i p_i x_i - \lambda \cdot \mathbb{E} \left( \sum_{i \in [n]} b_i Z_i - M \right)^2. \tag{3}$$

Expanding (3) yields a collection of terms which are constant, linear, and quadratic in $b_i$, and so the objective can be reformulated as an unconstrained binary quadratic program (UBQP). Although UBQP is strongly NP-hard, Çela et al. [3] present a pseudo-polynomial time algorithm for UBQP when the coefficient matrix for the quadratic form in the objective has constant rank. By proving that the objective (2) is sufficiently insensitive to small changes in our problem parameters, we leverage this pseudo-polynomial time algorithm to derive a FPTAS for approximating the optimal objective value for our problem. A standard search-to-decision reduction then yields the following result.

**Theorem 1.** *For $\rho = L_2$, Algorithm 4 identifies some $S \subseteq [n]$ satisfying $U(S) \geq U(S^*) - \epsilon$ in time $poly(1/\epsilon, n, M, \lambda)$.*

Pseudocode describing Algorithm 4 and the proof of this theorem are provided in Appendix B.5. On the other hand, by a reduction from equipartition we have the following hardness:

**Theorem 2.** *For $\rho = L_2$, optimizing $U(S)$ is weakly NP-hard.*

The proof of this theorem is provided in Appendix B.6. The hardness landscape of our problem when $\rho = L_2$ is therefore similar to that of the knapsack problem, which is heartening since the knapsack problem is relatively tractable in practice. However, unlike the knapsack problem, we should only hope for additive rather than multiplicative guarantees; this is because for $\rho = L_2$ our optimal value is not bounded away from zero and may be strictly negative, even if all values $x_i$ are nonnegative.

In contrast to $L_2$, we find that the one-sided loss $L_2^+$ is not straightforward to analyze. In this case the objective does not admit a quadratic factorization in terms of decision variables, and although the objective is nonnegative, it is difficult to analyze the performance of the greedy algorithm or contend with the nonlinearity of the loss function in a principled way. We instead turn to $L_1^+$ loss, where surprisingly these obstacles can be overcome.

### 3.2.2 Approximations for $L_1^+$ Loss

The loss $\rho = L_1^+$ enables the possibility of a multiplicative approximation because the optimum value $U(S^*)$ of the mixed-sign objective is nonnegative. More generally, any $\rho$ with $\rho(0, M) = 0$ has a nonnegative optimal value, since in this case $U(\emptyset) = 0$. For $\rho = L_1^+$, choosing any single item $i$ with positive $x_i$ and $p_i$ has strictly positive objective, since $M \geq 1$ implies that $L_1^+(1, M) = 0$ and so $U(\{i\}) = p_i x_i$. More generally, for this loss it suffices to consider only $i$ for which $x_i > 0$, since under this loss the marginal contribution of any $i$ with $x_i \leq 0$ is nonpositive.

The $L_1^+$ loss may appear amenable to a greedy algorithmic approach, since early items incur no penalty and the marginal penalty of adding a later item $i$ simply turns out to be proportional to the probability that the current solution exceeds the target $M$. However, as in the case for the $L_2$ loss, these natural heuristics fail to consider the relation between items in the selection. While efficient, these greedy algorithms perform arbitrarily badly compared to the optimal solution in the worst case. The failure of PGREEDY is again apparent as in the case for the $L_2$ loss. We now provide some informal "bad" instances for XGREEDY and XPGREEDY for intuition.

Recall that XGREEDY chooses items in decreasing order by value. Consider $M = 1$, and consider two types of items with $(x_1, p_1) = (1, p)$ and $(x_2, p_2) = (0.5, 1)$. XGREEDY picks item 1 and yields an objective of $O(p)$, whereas picking item 2 yields a constant objective. The other greedy algorithm XPGREEDY chooses items $i$ in decreasing order of their expected value $x_i p_i$. We consider the instance with $M = 1$, and two types of items $(x_1, p_1) = (1 + \epsilon, 1)$ and $(x_2, p_2) = (1/p, p)$ with some tiny $\epsilon > 0$ so that XPGREEDY chooses an item from type 1 and yields a constant objective. On the other hand, choosing $\frac{1}{p}$ copies of item 2 yields an objective of $\Omega(1/p)$.

For both XGREEDY and XPGREEDY, the constructed instances yield an upper bound on the approximation ratio, scaling as the minimum probability $p_{\min} := \min_{i \in [n]} p_i$ associated with the items. It suggests that $p_{\min}$ is a natural parameter for measuring the complexity of an instance with respect to the $L_1^+$ loss. Another natural parameter is $x_{\max} := \max_{i \in [n]} x_i$, the maximum value among all items.

Surprisingly, the performance of these greedy algorithms can be lower-bounded in terms of $p_{\min}$ as well, under an additional assumption about the values of items relative to $\lambda$.

**Theorem 3.** *Consider the one-sided loss $\rho = L_1^+$. If there is some fixed constant $c > 0$ such that $x_{max} \leq (1 - c) \cdot \lambda$, then*

    *(a) There exist instances for which PGREEDY selects $S$ with $U(S) = 0$, while $U(S^*) > 0$.*

    *(b) The worst-case approximation ratio for XPGREEDY is $\Theta(p_{min})$.*

    *(c) The worst-case approximation ratio for XGREEDY is $\Omega(p_{min}^2)$ and $O(p_{min})$.*

The proof of this theorem is provided in Appendix B.7. As a consequence, the approximation ratios of these greedy algorithms can be arbitrarily small as $p_{\min} \to 0$. Also note that the upper bounds do not require the assumption that $x_i \leq (1 - c)\lambda$; in all three cases there exist bad instances irrespective of this assumption. The reason we can derive better lower bounds for XPGREEDY than XGREEDY is intuitive; this is because XPGREEDY measures the expected reward conferred by an item, and so can be more directly related to the optimal solution $S^*$.

This is in notable contrast to when $\rho = L_2$, where we saw that multiplicative guarantees are inapt and all three greedy algorithms may incur arbitrarily large additive loss. Theorem 3 also raises a natural question: is this $p_{\min}$ upper bound tight, or is it possible to efficiently attain better approximations to $U(S^*)$ which do not depend on $p_{\min}$? We address this by introducing ONESIDEDL$_1^+$, presented in Algorithm 2, which attains a constant-factor approximation to the optimal solution. In this pseudocode, for any vector $v \in \mathbb{R}^n$ and set $S \subseteq [n]$, we use $v|_S$ to denote the $|S|$-dimensional vector obtained by restricting $v$ to its coordinates indexed by $S$. Its runtime is parameterized by the minimum probability $p_{\min}$ and the ratio of the maximum value $x_{\max}$ to the penalty parameter $\lambda$; in particular for fixed $p_{\min}$ and $\lambda/x_{\max}$ it is polynomial in $n$.

At a high level, ONESIDEDL$_1^+$ proceeds first by dividing the items into three groups according to their values $x_i$, and considering each group in turn. Since $U$ is submodular (see Lemma 1 in Appendix B.1), the optimal solution within at least one of these groups is also constant-competitive with $U(S^*)$. We obtain a constant-factor approximation for each group in a different way.

For the items with low values bounded away from $\lambda$, LOWVALUEL$_1^+$ (Algorithm 5 in Appendix B.8) checks all small subsets, which succeeds if the optimal subset for this group is small. It also computes rounded probabilities and values for each item in this group, and then efficiently computes the optimal solution according to this rounded instance. If the optimal subset is large, we then prove that this search over rounded solutions necessarily identifies a subset with objective value comparable to that of the optimal subset. This is the technical crux of proving that ONESIDEDL$_1^+$ is a constant-factor approximation.

For the items with values just below $\lambda$, MEDIUMVALUEL$_1^+$ (Algorithm 6 in Appendix B.8) returns the optimal subset if the group is small. If the group is large, it tries to choose a subset such that the expected number of realizations is about $M$; if there are not enough items, it chooses a subset with approximately half the expected number of realizations of the group overall. Finally, for the group of items with values above $\lambda$, it is straightforward to see that choosing the entire group is optimal. The pseudocode and related proofs for these algorithms appear in Appendix B.8. The following result provides a theoretical guarantee for Algorithm 2.

**Theorem 4** (Constant-factor approximation for $L_1^+$). *Algorithm 2 is a constant-factor approximation to $U(S^*)$ which runs in time $n^{O\left(\frac{1}{p_{min}^2} \max\left\{1, \log\left(\frac{1}{p_{min}}\right), \log\left(\frac{\lambda}{x_{max}}\right)\right\}\right)}$.*

The proof of this theorem is provided in Appendix B.8. Intuitively, the reason Algorithm 2 divides the items into cases depending on their values is to handle the case when the reward portion of $U(S^*)$ is almost equal to the penalty portion. This presents an impediment to the performance of solving a rounded version of the instance, since in this case the magnitude and even the sign of $U(S^*)$ is potentially quite sensitive to changes in $p_i$ and $x_i$. By restricting attention to items with low values, we prove that the expected number of realized items in $S^*$ is not much more than the target $M$. This then allows us to argue that there exist good rounded solutions that can be efficiently identified.

We conclude our theoretical results with a surprising equivalence between one- and two-sided linear losses $L_1^+$ and $L_1$. In what follows, we use $U_{L_1}$ and $U_{L_1^+}$ to denote the objective with $\rho = L_1$ and

---

**Algorithm 2** $\text{ONESIDEDL}_1^+$

---

**Require:** Problem instance $\mathcal{I} = (x, p, M, \lambda)$
**Ensure:** $S \subseteq [n]$ for which $U(S) \geq c \cdot U(S^*)$ for universal constant $c$
1: $N_L \leftarrow \{i \in [n] \colon x_i \leq (1 - \frac{p_{\min}}{4}) \cdot \lambda\}$
2: $N_M \leftarrow \{i \in [n] \colon (1 - \frac{p_{\min}}{4}) \cdot \lambda < x_i < \lambda\}$
3: $N_H \leftarrow \{i \in [n] \colon x_i \geq \lambda\}$
4: $S_L \leftarrow \text{LOWVALUEL}_1^+(x|_{N_L}, p|_{N_L}, \lambda, M)$
5: $S_M \leftarrow \text{MEDIUMVALUEL}_1^+(x|_{N_M}, p|_{N_M}, \lambda, M)$
6: $S_H \leftarrow N_H$
7: Compute $U(S_L)$, $U(S_M)$, and $U(S_H)$
8: **return** $S \in \{S_L, S_M, S_H\}$ maximizing $U(S)$

---

$\rho = L_1^+$, respectively. We use $U(S; \mathcal{I})$ to denote the evaluation of $U(S)$ specifically with respect to the instance $\mathcal{I} = (x, p, \lambda, M)$.

**Theorem 5** (Equivalence between $L_1$ and $L_1^+$). *For any instance $\mathcal{I} = (x, p, \lambda, M)$, construct $\mathcal{I}' = (x', p, \lambda', M)$ given by $x_i' := x_i - \lambda$ and $\lambda' := 2\lambda$. Then for all $S \subseteq [n]$,*

$$U_{L_1}(S; \mathcal{I}) = U_{L_1^+}(S; \mathcal{I}') - \lambda \cdot M.$$

The proof of this theorem is provided in Appendix B.9. In particular, since $\lambda$ and $M$ do not depend on $S$, this implies that $S$ maximizes $U_{L_1}$ on instance $\mathcal{I}$ if and only if it maximizes $U_{L_1^+}$ on instance $\mathcal{I}'$.

Although Theorem 5 establishes a correspondence between the solutions to our problem for $\rho = L_1$ and $\rho = L_1^+$, it is not approximation preserving, so it does not convert $\text{ONESIDEDL}_1^+$ into an approximation algorithm for the two-sided setting. Indeed, as in the $\rho = L_2$ setting, the optimal value when $\rho = L_1$ can be strictly negative.

## 4 Numerical Experiments

Having established worst-case theoretical guarantees, we wish to test how well our algorithms perform empirically. We focus on $L_1^+$ loss because our result for $L_2$ is an FPTAS, so we know its performance can be made arbitrarily close to optimal. Specifically, the experiments benchmark the subroutine $\text{LOWVALUEL}_1^+$ (part of $\text{ONESIDEDL}_1^+$) against XGREEDY and XPGREEDY for the regime where $x_i \leq (1 - c)\lambda$. This is the regime which $\text{LOWVALUEL}_1^+$ was developed to handle for $\text{ONESIDEDL}_1^+$, and it is the regime for which we prove performance guarantees for XPGREEDY and XGREEDY. In Appendix A, we also provide comparison to the optimal solution for smaller instances (Appendix A.1), and compare XGREEDY to XPGREEDY under other losses (Appendix A.2). All error bars shown in the plots represent standard error of the mean. The code to reproduce our simulation results is available at https://github.com/jingyanw/recruitment-uncertainty.

### 4.1 Experimental Setting

In constructing instances we follow the approach of Purohit et al. [12] in their use of beta distributions to orchestrate different kinds of correlation between $x_i$ and $p_i$. We therefore first draw $x_i \sim \text{Unif}[0, 1]$, and then produce three types of correlation as follows:

- *Negative correlation:* $p_i \sim p_{\min} + (1 - p_{\min}) \cdot \text{Beta}(10(1 - x_i), \ 10x_i)$.
- *Positive correlation:* $p_i \sim p_{\min} + (1 - p_{\min}) \cdot \text{Beta}(10x_i, \ 10(1 - x_i))$.
- *No correlation:* $p_i \sim \text{Unif}[p_{\min}, 1]$.

This construction differs from the sampling paradigm of Purohit et al. [12] only in that we re-normalize the probabilities $\{p_i\}$ so that they are bounded in $[p_{\min}, 1]$. We consider $n = 50$ and $p_{\min} = 0.01$ throughout, and explore the greedy heuristics XGREEDY and XPGREEDY, as well as the constant-factor approximation algorithm $\text{ONESIDEDL}_1^+$ (Algorithm 2), for a range of $M$ and $\lambda$.

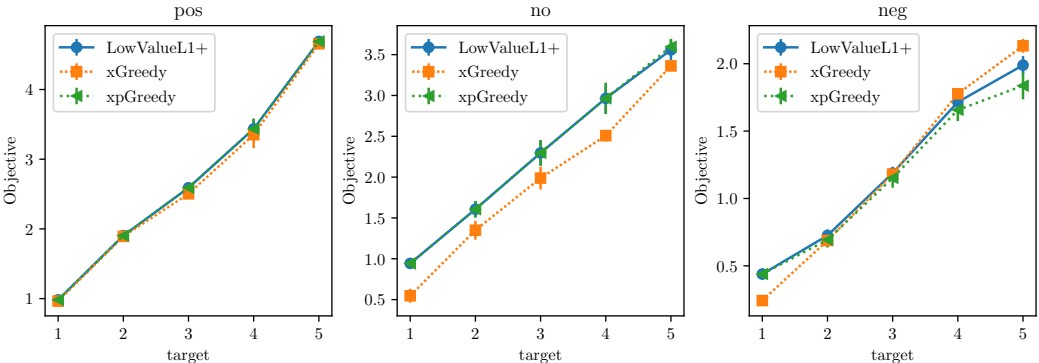

Figure 1: Sampling from the beta distribution with positive, no, and negative correlation. Here $n = 50$ and $\lambda = 3$.

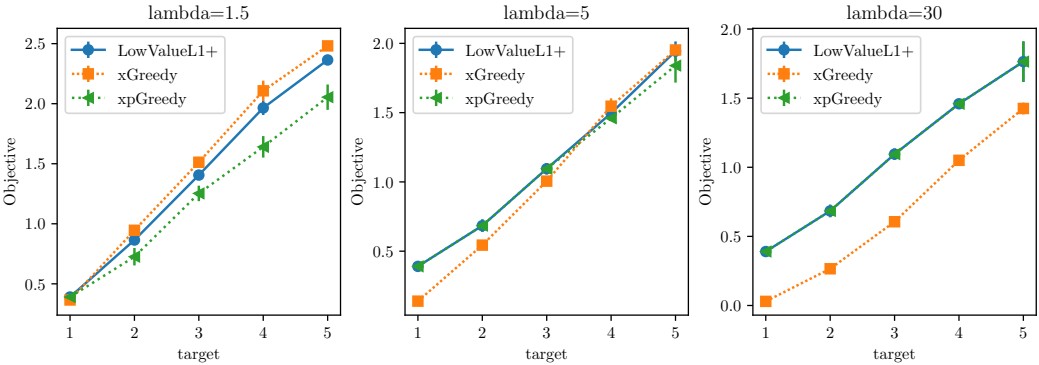

Figure 2: Performance for increasing penalty regularizer $\lambda$. Here $n = 50$ and sampling is via the negatively correlating beta distribution.

This lower bound on $p_{\min}$ ensures that the performance of the greedy heuristics and runtime of our algorithm are reasonable; a value of $0.01$ (say) is realistic because in practice, if a candidate takes the time and effort to apply, it is reasonable to assume that they at least have some nontrivial probability to accept if they were given an offer. We also focus on the regime where $x_i < \lambda$, which is assumed by Theorem 3 and handled in Algorithm 2 by the subroutine $\text{LowValueL}_1^+$. We believe that this is the main regime of practical interest: candidates with $x_i \geq \lambda$ are beneficial regardless of how many candidates have already accepted offers, and one might suppose that such candidates are rare.

Note that our theoretical guarantees in Theorem 4 necessitate that all candidate solutions up to size $\tau = \widetilde{O}(1/p_{\min}^2)$ are checked by brute force. In this implementation of $\text{LowValueL}_1^+$ we take $\tau = 0$ and isolate its search over rounded solutions. As we consider small target sizes $M$, this prevents $\text{LowValueL}_1^+$ from outperforming the greedy algorithms simply by virtue of having considered every relevant solution. This only hinders the performance of $\text{LowValueL}_1^+$. This implementation additionally only considers rounded solutions $S$ satisfying $\sum_{i \in S} x_i p_i \leq 2M$, which improves its runtime and theoretically only hinders its performance relative to $\text{LowValueL}_1^+$. So long as $x_i < \lambda$ and small solutions are checked, such a stopping condition can be implemented without hindering the performance of $\text{LowValueL}_1^+$; for details, see Appendix B.8 and Lemma 2 in Appendix B.1. We believe this provides a favorable tradeoff between runtime and accuracy, and illustrates a lower bound on the performance of $\text{LowValueL}_1^+$ as written.

## 4.2 Experimental Results

The objective values that our algorithms of interest attain for these distributions are shown in Figure 1. Note that positive correlation leads XGREEDY and XPGREEDY to pursue very similar (and optimal)

strategies, as expected. This is intuitively the easier setting, and here LOWVALUEL$_1^+$ performs on par with the greedy heuristics. In the no-correlation and negative-correlation settings, there are regimes where one of the two greedy heuristics performs better than the other one, whereas LOWVALUEL$_1^+$ appears to perform as well as the better of two depending on the regimes, showing its better adaptivity across these instances in practice as well as in theory.

Negative correlation between $x_i$ and $p_i$ is of particular interest to us, since it seems most relevant for the setting of faculty hiring and PhD admissions, and in fact hiring and recruitment more broadly. In Figure 1, we also observe that negative correlation is the setting that displays the most heterogeneity in algorithm behavior. We therefore turn to this negative-correlation setting and explore the effect of increasing the penalty regularizer $\lambda$ in Figure 2. In general, LOWVALUEL$_1^+$ appears comparable to the better of the two greedy heuristics across the values of $M$ and $\lambda$ that we examine, though there is a small gap between the objectives achieved by LOWVALUEL$_1^+$ and XGREEDY when $\lambda = 1.5$.

This is also good news for XGREEDY and XPGREEDY, because it suggests that the two of them together remain competitive across a wide range of instances. To the extent that LOWVALUEL$_1^+$ falls short of the objective achieved, it is due to systematically rounding the probabilities $p_i$ up by a constant factor when computing the prospective utility of solutions. Because its rounding preserves the reward term, such a systematic overestimate in $p_i$ leads it to overestimate the penalty term of any set under consideration. The impact of rounding may be small on each individual item but collectively large on the objective, and therefore explains the extent to which LOWVALUEL$_1^+$ lags behind XGREEDY in Figure 2; the latter chooses many such items while the former judges their influence on the penalty to be too large. However, this can be mitigated by choosing smaller multiplicative bucket sizes for LOWVALUEL$_1^+$ in rounding, which is particularly effective in the case where the probabilities $\{p_i\}$ of an instance fall in a small number of clusters or exhibit other structure.

## 5    Discussion

One of the takeaways from our theoretical and empirical results is that, in addition to XPGREEDY, the greedy algorithm XGREEDY, which makes offers to a subset of candidates with the highest values, is practicable for $L_1^+$ loss. This is intriguing because the algorithm is quite similar to how faculty hiring and admissions committees typically think: they want to make offers to the best candidates. The difference is that XGREEDY carefully selects the *number* of offers to be made, in a way that greedily maximizes the objective. Since XGREEDY amounts to a relatively small tweak to current practice, we believe committees would find the algorithm to be especially palatable.

An issue our results do not address is which penalty function best matches the needs of a specific recruitment process. For example, is there a rigorous way to argue that a particular choice of penalty function is more broadly applicable than another? That said, the choice between one-sided and two-sided penalty is rather intuitive, depending on the application. And our results provide computational arguments in favor of $L_2$ when two-sided penalty is desired, and $L_1^+$ for one-sided penalty.

From an ethical viewpoint, a potential concern is that our proposal may ultimately have unintended negative consequences. For example, if many faculty hiring committees adopted our optimization-based approach, might candidates have fewer opportunities? We believe, however, that the opposite is true. Currently the academic job market is strikingly inefficient, as committees often converge on a few candidates who are inundated with interviews and offers, while comparably strong candidates are left with nothing. If our approach is adopted widely, it is likely to widen the pool of candidates who receive appealing offers. Granted, a centralized matching market (in the style of the National Resident Matching Program) may be an even better solution, but creating such a market requires a huge — and often impractical — degree of coordination; by contrast, our approach can be adopted independently by institutions and even by individual departments or committees.

**Acknowledgments.** We thank Anupam Gupta and Alejandro Toriello for helpful discussions. GK and AP were partially supported by the National Science Foundation under grants IIS-2147187, CCF-2007080, IIS-2024287, and CCF-1733556; and by the Office of Naval Research under grant N00014-20-1-2488. JW was supported by the Ronald J. and Carol T. Beerman President's Postdoctoral Fellowship and the ARC (Algorithms & Randomness Center) Postdoctoral Fellowship at Georgia Tech.

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
