# A  Additional Experiments

In this section, we present additional experiments which shed more light on the performance of XGREEDY, XPGREEDY, and ONESIDEDL$_1^+$ relative to one another and to the optimal solution, for a broader range of objectives. The family of distributions from which we sample instances is the same as the one described in Section 4.1.

## A.1  Comparison to Optimal

First, we recreate Figure 1 and Figure 2, now including the objective value of the optimal solution $S^*$ as a benchmark for the three algorithms considered above. Since determining $U(S^*)$ by brute force is computationally costly, this comparison is undertaken for smaller instances ($n = 20$). Following Section 4.1, we consider $\lambda = 3$.

Here Figure 3 shows the performance of XGREEDY, XPGREEDY, and ONESIDEDL$_1^+$ relative to the objective $U(S^*)$ of the optimal solution, when values and probabilities are positively correlated, uncorrelated, and negatively correlated, for a range of target sizes $M$. Figure 4 shows the performance of XGREEDY, XPGREEDY, and ONESIDEDL$_1^+$ relative to $U(S^*)$ as the penalty regularizer increases, for negatively correlated $x_i$ and $p_i$ and again for a range of target sizes $M$.

It is noteworthy that in both Figure 3 and Figure 4, the best algorithms in each setting nearly attain the optimal objective value. It is unclear the extent to which we should expect that this continues to hold for larger instances, where solving the optimal solution by brute force is computationally infeasible.

## A.2  Other Objectives

In Section 4.2 and Appendix A.1, we examine the performance of different algorithms for the $L_1^+$ loss, since this is the loss function for which we derive worst-case multiplicative guarantees and for which the algorithm ONESIDEDL$_1^+$ was designed.

We now investigate how these algorithms perform with respect to other loss functions, despite the absence of worst-case theoretical guarantees for the greedy heuristics. Figure 5 compares the greedy heuristics between the $L_1^+$ and $L_2^+$ loss objectives across different correlation regimes. Figure 6 does the same for the two-sided $L_1$ and $L_2$ loss objectives.

In Figure 5 the performance of both greedy heuristics is very similar under the two one-sided losses. For the two-sided losses $L_1$ and $L_2$, Figure 6 suggests that XPGREEDY dramatically outperforms XGREEDY across the choice of the two-sided losses. We observe that the objective values are no longer uniformly positive, and are no longer monotonically increasing in the target size. This is because the problem under the two-sided losses is fundamentally more difficult: under one-sided losses, only selecting over the target is penalized; it is straightforward to observe that selecting $M$ items always yields a penalty of 0 and hence a positive objective value. Under two-sided losses, selecting under the target and selecting over the target is both penalized; there is also non-zero variance

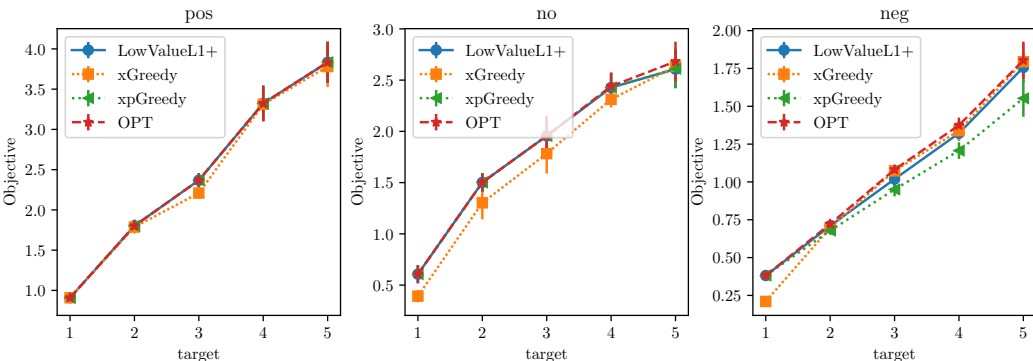

Figure 3: Sampling from the beta distribution with positive, no, and negative correlation. Here $n = 20$ and $\lambda = 3$, and OPT denotes $U(S^*)$.

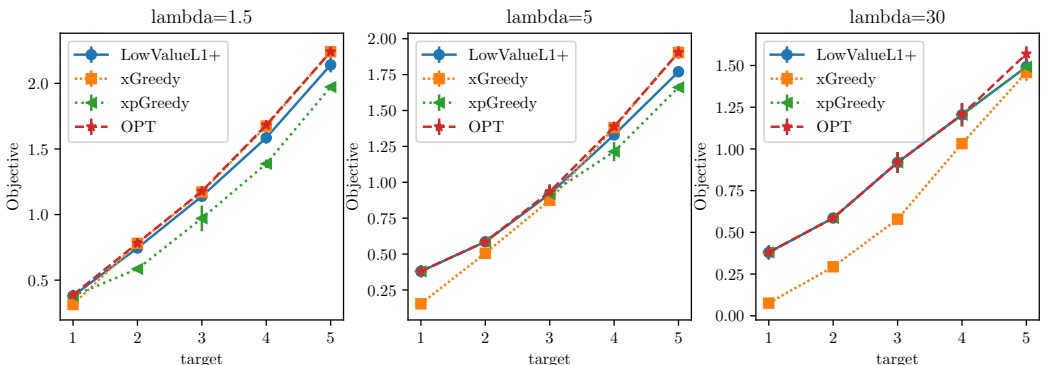

Figure 4: Performance for increasing penalty regularizer $\lambda$. Here $n = 20$ and sampling is via the negatively correlating beta distribution.

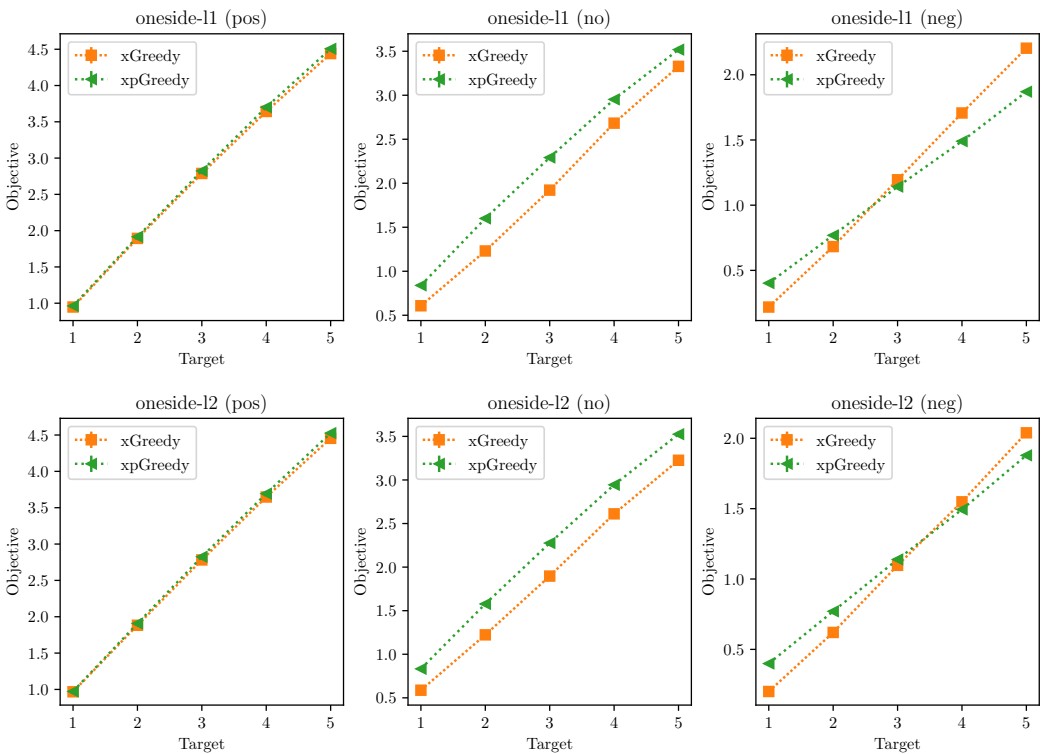

Figure 5: Evaluation of greedy heuristics for $L_1^+$ versus $L_2^+$ one-sided loss. Here $n = 50$ and $\lambda = 3$.

towards achieving the exact target $M$, and hence the objective is negative when the regularizer $\lambda$ is large.

Comparing the two-sided losses $L_1$ and $L_2$ in Figure 6, the problem under the $L_2$ loss is more difficult due to its higher penalty (the quadratic function always attains a higher value than the linear function on integers). The objective starts decreasing as a function of the target $M$: if we are aiming at a larger target $M$, more items are selected, leading to an inevitable increase in the variance and hence a lower objective.

We provide an informal explanation for the superior performance of XPGREEDY over XGREEDY for the two-sided losses, using the two-sided $L_2$ loss as an example. Under this loss, a candidate $i$ contributes $x_i p_i$ to the reward term of the objective, while contributing $p_i(1 - p_i)$ to the variance of the realized size. When faced with two candidates of equal value $x_i$, we should therefore at the

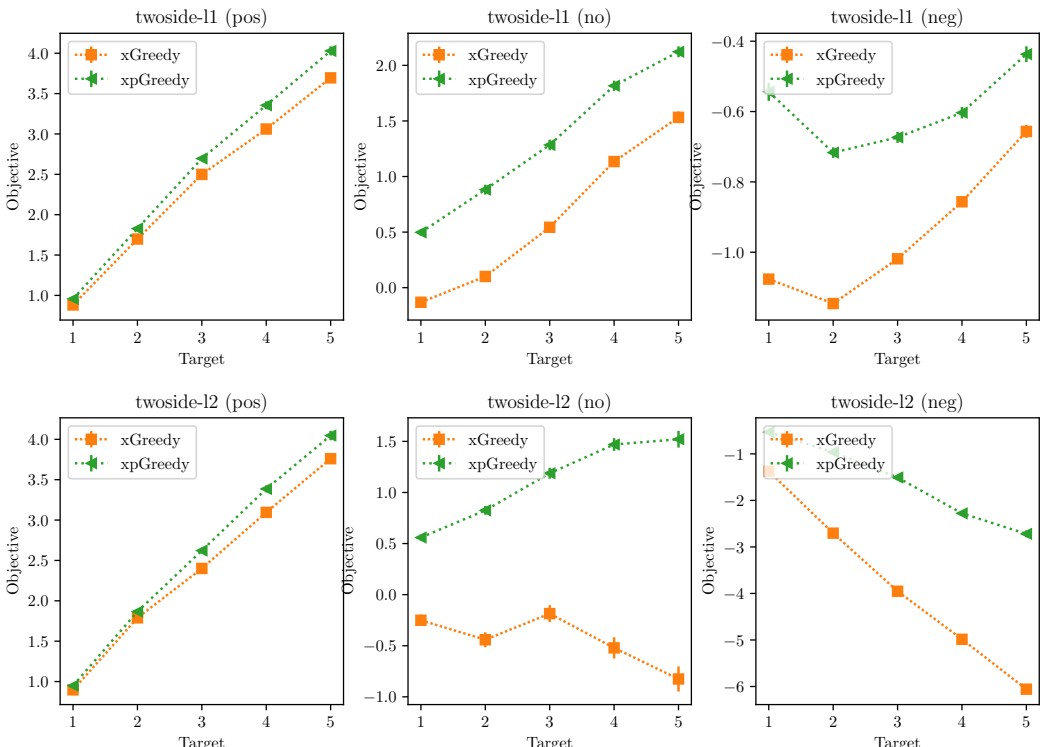

Figure 6: Evaluation of greedy heuristics for $L_1$ versus $L_2$ two-sided loss. Here $n = 50$ and $\lambda = 3$.

margin prefer the candidate with the higher probability, since this candidate contributes less to the variance per contribution to the reward. Note that for sufficiently large $\lambda$, two-sided losses encourage algorithms to choose solutions whose expected size is very close $M$, meaning that the variance and the penalty term are nearly equal. Here XPGREEDY prefers this higher-probability candidate, while XGREEDY is indifferent, explaining the superior performance of XPGREEDY.

## B  Proofs

In this section, we present the proofs of all theoretical results.

### B.1  Preliminaries

For any set or event $S$, we use $\overline{S}$ to denote its complement. We use the notation $f(x) \lesssim g(x)$ to denote that there exists some universal positive constant $c > 0$, such that $f(x) \leq c \cdot g(x)$, and use the notation $f(x) \gtrsim g(x)$ when $g(x) \lesssim f(x)$.

For any vector $x \in \mathbb{R}^n$ and set $S \subseteq [n]$, we use the shorthand $x_S := \{x_i\}_{i \in S}$. Let $\mu_S := \mathbb{E}[\sum_{i \in S} Z_i] = \sum_{i \in S} p_i$. We also denote by $\mu^* := \mu_{S^*}$ the expected size of the optimal subset.

The following lemma shows the submodularity of the objective $U$ in the selection $S$.

**Lemma 1.** *If $\rho(\cdot, M)$ is convex then $U(S)$ is submodular in $S$.*

The proof of this lemma is provided in Appendix B.2. The submodularity is used in the proof of Theorem 4 (in Appendix B.8).

For the $L_1^+$ loss, the following lemma allows us to reason about the cardinality of $S^*$ in the case when the penalty $\lambda$ is larger than any of the item values.

**Lemma 2** (Mean Bound). *Consider the $L_1^+$ objective. There exists a universal constant $c_0 > 0$ such that the following is true. For any $\epsilon \in (0, \frac{3}{4})$, if $x_{max} \leq (1 - \epsilon) \cdot \lambda$, then either*

$$|S^*| \leq \frac{c_0 \log(\frac{1}{\epsilon})}{p_{min}} \tag{4a}$$

*or*

$$\mu^* \leq \frac{101}{100} M. \tag{4b}$$

The proof of this lemma is provided in Appendix B.2.1. It is used in the proofs of Theorem 3 and Theorem 4. Intuitively, this lemma says that when all values are less than and bounded away from $\lambda$, one of the following two caeses is true: either the number of items in the optimal solution is small (Eq. (4a)), or when the number of items in the optimal solution is large, then the realized size of the optimal solution concentrates to the expected size with a small variance. Since the item values are relatively small, in order to minimize the $L_1^+$ loss, the expected size is at most on the order of $M$ (Eq. (4b)).

## B.2 Proof of Lemma 1

We write out the objective $U(S)$ over all possible realizations of $Z \in \{0,1\}^n$ as:

$$U(S) := R(S) - \lambda \cdot \mathbb{E}\left[\rho\left(|S_Z|,\ M\right)\right]$$

$$= \sum_{i \in S} p_i x_i - \lambda \sum_{z \in \{0,1\}^n} \mathbb{P}(Z = z) \cdot \rho\left(\sum_{i \in S} z_i,\ M\right). \tag{5}$$

The first term in (5) is additive, and hence submodular. Since $\rho$ is convex, it can be verified that the loss $\rho\left(\sum_{i \in S} z_i,\ M\right)$ is supermodular in $S$ for each fixed realization $z$. Taking linear combinations of these terms yields the submodularity of $U(S)$. $\qquad\square$

### B.2.1 Proof of Lemma 2

Recall that $S^*$ denotes the optimal solution under the $L_1^+$ objective. Denote by $i^* := \arg\min_{i \in S^*} p_i$ the item in the optimal selection with the minimal probability, and denote $S := S^* \setminus \{i^*\}$. In what follows, we prove claim (4) by deriving a lower bound and an upper bound on $\mathbb{P}(\sum_{i \in S} Z_i \leq M)$.

**Lower bounding $\mathbb{P}(\sum_{i \in S} Z_i \leq M)$ by the optimality of $S^*$.** Recall that the $L_1^+$ loss penalizes the case when the total number of accepted items exceeds $M$. Intuitively, adding item $i^*$ to the set $S$ is only beneficial if $\mathbb{P}(\sum_{i \in S} Z_i \geq M)$ is small. Formally, we have

$$U(S^*) - U(S) = x_{i^*} p_{i^*} - \lambda \mathbb{E}\left[(\sum_{i \in S} Z_i + Z_{i^*} - M)_+ - (\sum_{i \in S} Z_i - M)_+\right]$$

$$= x_{i^*} p_{i^*} - \lambda p_{i^*} \cdot \mathbb{P}\left(\sum_{i \in S} Z_i \geq M\right)$$

By the optimality of $S^*$, we have $U(S^*) \geq U(S)$, and hence

$$\mathbb{P}\left(\sum_{i \in S} Z_i \geq M\right) \leq \frac{x_{i^*}}{\lambda} \overset{(i)}{\leq} 1 - \epsilon,$$

where step (i) is true by the assumption that $x_{\max} \leq (1 - \epsilon)\lambda$. Hence, we have

$$\mathbb{P}\left(\sum_{i \in S} Z_i \leq M\right) \geq \mathbb{P}\left(\sum_{i \in S} Z_i < M\right) \geq \epsilon. \tag{6}$$

**Upper bounding $\mathbb{P}(\sum_{i \in S} Z_i \leq M)$ by concentration.** Let the universal constant $c_0$ satisfy $c_0 \geq \frac{200}{\log(\frac{4}{3})}$. If $|S^*| < 200$, then we have

$$|S^*| < 200 \leq \frac{c_0 \log(\frac{1}{\epsilon})}{p_{\min}},$$

satisfying (4a). Hence, it remains to consider the case when $|S^*| \geq 200$. In what follows, we assume that condition (4b) does not hold. That is, we assume $\mu^* > \frac{101}{100} M$. Then we prove that condition (4a) holds. We derive a multiplicative Chernoff bound to upper bound $\mathbb{P}(\sum_{i \in S} Z_i \leq M)$. We first establish a relation between $\mu_S$ and $M$. Using the definition that $i^*$ is the item with the smallest probability in the optimal selection $S^*$, we have

$$\mu^* = \sum_{i \in S} p_i + p_{i^*} \leq \frac{|S^*|}{|S^*| - 1} \mu_S \overset{(i)}{\leq} \frac{200}{199} \mu_S, \tag{7}$$

where step (i) uses the assumption that $|S^*| \geq 200$. Combining (7) with the assumption that condition (4b) does not hold and hence $\mu^* > \frac{101}{100} M$, we have

$$\frac{101}{100} M < \mu^* \leq \frac{200}{199} \mu_S$$
$$M < \frac{200}{199} \cdot \frac{100}{101} \mu_S \leq (1 - c)\mu_S,$$

where $c > 0$ is a universal constant. Since $\mu_S > M$, by the multiplicative Chernoff bound,

$$\mathbb{P}\Big(\sum_{i \in S} Z_i \leq M\Big) \leq \mathbb{P}\Big(\sum_{i \in S} Z_i \leq (1 - c)\,\mu_S\Big) \leq \mathrm{Exp}\Big(-\frac{c^2 \mu_S}{2}\Big) \leq \mathrm{Exp}\Big(-\frac{c^2 \cdot |S| \cdot p_{\min}}{2}\Big). \tag{8}$$

**Combining the lower and the upper bounds.** Combining the lower bound (6) and the upper bound (8) on $\mathbb{P}(\sum_{i \in S} Z_i \leq M)$, we have

$$\epsilon \leq \mathbb{P}\Big(\sum_{i \in S} Z_i \leq M\Big) \leq \mathrm{Exp}\Big(-\frac{c^2 \cdot |S| \cdot p_{\min}}{2}\Big)$$

and so

$$|S| \leq \frac{2}{c^2} \cdot \frac{\log(\frac{1}{\epsilon})}{p_{\min}}.$$

By the assumption that $\epsilon \leq \frac{3}{4}$, we have $|S^*| = |S| + 1 \leq \frac{c_0 \log(\frac{1}{\epsilon})}{p_{\min}}$ for some universal constant $c_0 > 0$, satisfying condition (4a). $\qquad \square$

### B.3   Proof of Proposition 1

We fix any target size $M > 0$. For notational simplicity, we use the shorthand $\rho(\cdot) := \rho(\cdot, M)$ for the loss function. For the set $\{1, 2, \ldots, n\}$, we say that the subset $\{1, 2, \ldots, k\}$ for each $k \in \{0, \ldots, n\}$ is a prefix. Recall that Algorithm 1 sorts the items in decreasing order of probability as $p_1 \geq \ldots \geq p_n$, with ties broken arbitrarily. In what follows, we first show that there exists a prefix of items that is an optimal selection. Then we show that using the stopping criterion in Line 4 of Algorithm 1 achieves the minimum variance among all prefixes, and hence is an optimal selection.

**Showing that a prefix of items in decreasing order of $p_i$ achieves an optimal selection.** Assume that there exists an optimal selection, denoted by $S^* \subseteq [n]$, that is not a prefix in decreasing order of $p_i$. By the assumption that $S^*$ is not a prefix of items in decreasing order of $p_i$, there must exist items $i \in S^*$ and $i' \notin S^*$, such that $p_{i'} \geq p_i$. We now show that $S^{*\prime} := S^* \cup \{i'\} \setminus \{i\}$, namely removing $i$ from $S^*$ and then adding $i'$, also yields an optimal selection.

If $p_i' = p_i$, it is straightforward to see that the variance remains the same, and hence $S^{*\prime} = S^* \cup \{i'\} \setminus \{i\}$ is optimal. Now we consider the case $p_{i'} > p_i$. For any subset $S \subseteq [n]$, we consider

the additional variance induced by adding any item $k \notin S$ to the subset $S$:

$$V\left(S \cup \{k\}\right) - V\left(S\right) = \mathbb{E}_{Z_{S \cup \{k\}}}\left[\rho\left(\sum_{i \in S \cup \{k\}} Z_i\right) - \rho\left(\sum_{i \in S} Z_i\right)\right]$$

$$\overset{(i)}{=} \mathbb{E}_{Z_S}\left[\mathbb{E}_{Z_k}\,\rho\left(\sum_{i \in S \cup \{k\}} Z_i\right) - \rho\left(\sum_{i \in S} Z_i\right)\right]$$

$$\overset{(ii)}{=} p_k \cdot \underbrace{\mathbb{E}_{Z_S}\left[\rho\left(\sum_{i \in S} Z_i + 1\right) - \rho\left(\sum_{i \in S} Z_i\right)\right]}_{T(S)}, \qquad (9)$$

where (i) is true by the assumption that the random variables $\{Z_i\}_{i=1}^n$ are independent, and (ii) is true by taking an expectation over $Z_k$. Setting $S = S^* \setminus \{i\}$ and $k \in \{i, i'\}$ in (9), we have

$$V\left(S^*\right) - V(S^* \setminus \{i\}) = p_i \cdot T(S^* \setminus \{i\}), \qquad (10a)$$

$$V\left(S^{*\prime}\right) - V(S^* \setminus \{i\}) = p_{i'} \cdot T(S^* \setminus \{i\}), \qquad (10b)$$

Combining (10a) with the assumption that $S^*$ is an optimal selection, we have $T(S^* \setminus \{i\}) \geq 0$. Combining (10) with the assumption that $p_{i'} > p_i$, we have

$$V\left(S^{*\prime}\right) \geq V\left(S^*\right). \qquad (11)$$

Since by assumption $S^*$ is an optimal selection, equality holds in (11) and $S^{*\prime}$ is also an optimal selection.

If $S^{*\prime}$ is not a prefix, we keep repeating the same modification, until the resulting selection is a prefix. Since $p_{i'} \geq p_i$, we have $i' \leq i$, and hence in each modification, the sum of the indices in the selection decreases, namely $\sum_{k \in S^{*\prime}} k < \sum_{k \in S^*} k$. Hence, the sequence of modifications terminates, yielding an optimal selection that is a prefix.

**Showing that the stopping criterion obtains a best prefix among all prefixes.** We now show that the stopping criterion in Line 4 of Algorithm 1 obtains a prefix with the minimum variance among all prefixes. Since we have showed that there exists a prefix that is an optimal selection, this prefix obtained by the stopping criterion is optimal.

We consider the term $T$ in (9) when adding to a selection $S \subseteq [n]$ some new item $k \notin S$. We have

$$T(S \cup \{k\}) = \mathbb{E}_{Z_S}\mathbb{E}_{Z_k}\left[\rho\left(\sum_{i \in S \cup \{k\}} Z_i + 1\right) - \rho\left(\sum_{i \in S \cup \{k\}} Z_i\right)\right]$$

$$\overset{(i)}{=} p_k\mathbb{E}_{Z_S}\left[\rho\left(\sum_{i \in S} Z_i + 2\right) - \rho\left(\sum_{i \in S} Z_i + 1\right)\right] + (1 - p_k) \cdot \mathbb{E}_{Z_S}\left[\rho\left(\sum_{i \in S} Z_i + 1\right) - \rho\left(\sum_{i \in S} Z_i\right)\right]$$

$$\overset{(ii)}{\geq} \mathbb{E}_{Z_S}\left[\rho\left(\sum_{i \in S} Z_i + 1\right) - \rho\left(\sum_{i \in S} Z_i\right)\right] = T(S), \qquad (12)$$

where step (i) takes an expectation over $Z_k$, and step (ii) uses the property that $\rho(t+2) - \rho(t+1) \geq \rho(t+1) - \rho(t)$ for any $t \in \mathbb{R}$, due to the convexity of $\rho$. Due to the stopping criterion, Algorithm 1 yields a prefix $\{1, 2, \ldots, i^*\}$ such that $T([i]) \leq 0$ for all $i \leq i^*$, and $T([i^* + 1]) > 0$. By (12), it can be verified that $T([i]) > 0$ for all $i > i^*$. Hence, the variance decreases or stays the same for adding each item up to item $i^*$, and then strictly increases for adding each of item $(i^* + 1)$ through item $n$. Hence, the prefix $[i^*]$ attains the minimal variance among all prefixes, and hence is an optimal selection. $\qquad\square$

### B.4 Proof of Proposition 2

Consider any instance $(x, p, \lambda, M)$ and any constant $c > 0$. It is straightforward to verify that the optimal solution and the solution given by any of the three greedy algorithms is identical for the instance $(x, p, \lambda, M)$ and the instance $(cx, p, c\lambda, M)$. Hence, it suffices to construct an instance for a fixed value of $\lambda > 0$. We now construct instances for the greedy algorithms separately.

**Instance for PGREEDY.** Let $M = 1$. We consider an instance consisting of two items:
$$(x_1, p_1) = (0, 1)$$
$$(x_2, p_2) = (1, p),$$
for some $p \in (0, 1)$ whose value is specified later. It is straightforward to derive that PGREEDY selects item 1, attaining an objective of $0$. On the other hand, the objective of only picking item 2 is:
$$p - \lambda(1 - p).$$
We take $p$ to be sufficiently close to 1 such that $p/(1 - p) > \lambda$. Then the objective of only picking item 2 is strictly positive, and hence the objective of the optimal solution is strictly positive.

**Instance for XGREEDY and XPGREEDY.** Let $M = 1$. We consider an instance consisting of two items:
$$(x_1, p_1) = (1, 1)$$
$$(x_2, p_2) = \left(2 + \epsilon, \frac{1}{2}\right),$$
for some $\epsilon > 0$ whose value is specified later. The objective for the four possible selections is computed as:
$$U(\emptyset) = -\lambda$$
$$U(\{1\}) = 1$$
$$U(\{2\}) = 1 + \frac{\epsilon - \lambda}{2}$$
$$U(\{1, 2\}) = 2 + \frac{\epsilon - \lambda}{2}.$$
It is straightforward to derive that both XGREEDY and XPGREEDY pick item 2 first followed by item 1, attaining an objective of $2 + \frac{\epsilon - \lambda}{2}$. We set any value of $\lambda$ such that $\lambda > 4$, and set $\epsilon = \frac{\lambda}{2} - 2 > 0$. The objective becomes $1 - \frac{\lambda}{4} < 0$. On the other hand, the optimal selection is $S^* = \{1\}$, with a strictly positive objective of $1$. $\qquad\square$

### B.5 Proof of Theorem 1

We first describe a pseudo-polynomial time algorithm proposed by Çela et al. [3] for solving a specific form of rank-1 binary quadratic programming. Then we describe our algorithm, which operates by rounding the parameters and using the pseudo-polynomial time algorithm as a sub-routine.

**Pseudo-polynomial time algorithm of [3].** Çela et al. [3] study unconstrained binary quadratic programming problems of the form
$$\min_{x \in \{0,1\}^n} \langle x, Ax \rangle + \langle b, x \rangle.$$
where $A \in \mathbb{R}^{n \times n}$ is symmetric and $a \in \mathbb{R}^n$. When $A$ has rank one, this can be reformulated as
$$\min_{x \in \{0,1\}^n} \langle a, x \rangle + \gamma(\beta + \langle u, x \rangle)^2 \tag{13}$$
for some $a, u \in \mathbb{R}^n$ and $\gamma, \beta \in \mathbb{R}$. We note that the representation $(a, u, \gamma, \beta)$ for the problem (13) is not unique. Çela et al. [3] propose an algorithm to solve (13) exactly with a run time dependent on the magnitude of the representation.

**Proposition 3** (Proposition 1 of [3] with $d = 1$)**.** *Consider any instance of* (13) *with* $u \in \mathbb{Z}^n$, $\beta \in \mathbb{Z}$, $a \in \mathbb{Z}^n$, *and* $\gamma \in \mathbb{Q}$. *Let* $K := 2 \max(\|u\|_\infty, \|a\|_\infty)$. *Then the minimum objective attained by* (13) *can be computed in* $O(K^4 n^5)$ *time.*

We refer to the algorithm satisfying Proposition 3 as R1UBQPSOLVER, which is described in the proof of Proposition 1 in Çela et al. [3]. R1UBQPSOLVER takes as inputs $(a, u, \gamma, \beta)$, and outputs the minimum objective attained by (13). We also note that while R1UBQPSOLVER as stated requires $\gamma \in \mathbb{Q}$, this serves only as a sufficient condition for arguing that arithmetic operations involving $\gamma$ can be performed efficiently. Since our runtime analysis is in terms of the number of arithmetic operations performed, we remain agnostic to the exact representation of the numbers in our problem instance.

**Modified R1UBQP Solver.** For convenience of the presentation, we use a slightly more general version of this binary quadratic programming algorithm, which essentially serves as a rescaling of R1UBQPSOLVER. We will call this R1UBQPSOLVER2 (Algorithm 3).

---

**Algorithm 3** R1UBQPSOLVER2

---

**Require:** $u \in \mathbb{Z}^n$, $\beta \in \mathbb{Z}$, $a \in \mathbb{Q}^n$ such that $a = \frac{a'}{E}$ for some $a' \in \mathbb{Z}^n$ and $E \in \mathbb{N}$, and $\gamma \in \mathbb{R}$
**Ensure:** $\min_{b \in \{0,1\}^n} \langle a, b \rangle + \gamma(\beta + \langle u, b \rangle)^2$
 1: $a' \leftarrow Ea$
 2: $\gamma' \leftarrow E\gamma$
 3: **return** $\frac{1}{E}$R1UBQPSOLVER$(a', \gamma', \beta, u)$

---

**Proposition 4.** *Consider an instance of* (13) *with* $u \in \mathbb{Z}^n$, $\beta \in \mathbb{Z}$, $a \in \mathbb{Q}^n$ *such that* $a = \frac{a'}{E}$ *for some* $a' \in \mathbb{Z}^n$ *and* $E \in \mathbb{N}$, *and* $\gamma \in \mathbb{R}$. *Let* $K' := 2\max(\|u\|_\infty, \|a'\|_\infty)$. *Then* R1UBQPSOLVER2 *computes the minimum objective attained by* (13) *in* $O(K'^4 E^4 n^5)$ *time.*

*Proof.* The correctness of R1UBQPSOLVER2 follows immediately from Proposition 3. For the runtime guarantee, note that the invocation of R1UBQPSOLVER is with $a'_i = Ea_i$, so the guarantee from R1UBQPSOLVER holds with $K \geq EK'$. $\qquad\square$

**Proposed FPTAS.** We now derive the following FPTAS for our problem, which uses R1UBQPSOLVER2 as a subroutine:

---

**Algorithm 4** APPROXL$_2$

---

**Require:** Problem instance $\mathcal{I} = (x, p, \lambda, M)$; additive error $\epsilon > 0$
**Ensure:** $S \subseteq [n]$ for which $U(S) \geq U(S^*) - \epsilon$
 1: $D \leftarrow \lceil 2n\lambda(2M + 3(n+1))/\epsilon \rceil$
 2: $E \leftarrow \lceil 2n/\epsilon \rceil$
 3: $\bar{a} \leftarrow \frac{1}{E}\lfloor E(-p \circ x + \lambda \cdot p - \lambda \cdot (p \circ p)) \rfloor$
 4: $\gamma' \leftarrow \lambda/D^2$
 5: $\beta' \leftarrow DM$
 6: $u' \leftarrow \lfloor Dp \rfloor$
 7: $\overline{OPT} \leftarrow -$R1UBQPSOLVER2$(\bar{a}, \gamma', \beta', u')$
 8: $S \leftarrow [n]$
 9: **while** $\exists i \in S$ such that $-$R1UBQPSOLVER2$(\bar{a}|_{S\setminus\{i\}}, \gamma'|_{S\setminus\{i\}}, \beta'|_{S\setminus\{i\}}, u'|_{S\setminus\{i\}}) = \overline{OPT}$ **do**
10: $\quad\quad S \leftarrow S \setminus \{i\}$
11: **return** $S$

---

Given an instance of our problem, APPROXL$_2$ generates a rounded instance and then runs R1UBQPSOLVER2 on this rounded instance. The objective is guaranteed to be close to optimal, and so APPROXL$_2$ first finds the optimal rounded value, and then identifies a set which attains this rounded value.

We prove that APPROXL$_2$ (Algorithm 4) is a FPTAS for our problem when $\rho = L_2$.

**Rewriting the objective in the form of** (13). We begin by establishing that if $\rho = L_2$ then $U(S)$ can be written in the form of Eq. (13). Recall from (2) that our objective can be written as

$$U(S) = \sum_{i \in S} p_i x_i - \lambda \cdot \mathbb{E}\left(\sum_{i \in S} Z_i - M\right)^2.$$

Recall from (3) the notation $b \in \{0,1\}^n$ with $b_i := \mathbb{1}\{i \in S\}$. The vector $b$ is a representation of the set $S$, and we slightly abuse notation to let $U(b) := U(S)$. We have

$$U(b) = \sum_{i \in [n]} b_i p_i x_i - \lambda \cdot \mathbb{E}\left(\sum_{i \in [n]} b_i Z_i - M\right)^2. \tag{14}$$

For any two vectors $u, v \in \mathbb{R}^n$, let $u \circ v$ denote the entrywise product. Expanding the squared term in (14) yields

$$U(b) = \sum_{i \in [n]} b_i p_i x_i - \lambda \cdot \left( \mathbb{E}\left( \sum_{i \in [n]} b_i Z_i \right)^2 - 2M \sum_{i \in [n]} b_i p_i + M^2 \right)$$

$$\overset{(i)}{=} (p \circ x)^T b - \lambda \cdot \mathbb{E}\left[ \sum_{i \in [n]} b_i Z_i + \sum_{(i,j) \in [n]^2, i \neq j} b_i b_j Z_i Z_j \right] + 2\lambda M \cdot p^T b - \lambda M^2$$

$$= (p \circ x)^T b - \lambda \cdot p^T b - \lambda \cdot \sum_{(i,j) \in [n]^2, i \neq j} b_i b_j p_i p_j + 2\lambda M \cdot p^T b - \lambda M^2$$

$$= (p \circ x)^T b - \lambda \cdot p^T b - \lambda \cdot \left( \left( \sum_{i \in [n]} b_i p_i \right)^2 - \sum_i b_i^2 p_i^2 \right) + 2\lambda M \cdot p^T b - \lambda M^2,$$

where step (i) uses the fact that $b_i$ and $Z_i$ are binary, and hence $b_i^2 = b_i$ and $Z_i^2 = Z_i$. Using the fact that $b_i$ is binary again, we have

$$U(b) = (p \circ x)^T b - \lambda \cdot p^T b - \lambda \cdot (p^T b)^2 + \lambda \cdot (p \circ p)^T b + 2\lambda M \cdot p^T b - \lambda M^2$$

$$= (p \circ x - \lambda \cdot p + \lambda \cdot (p \circ p))^T b - \lambda \left( -M + p^T b \right)^2. \tag{15}$$

Negating (15) yields

$$\min_{S \subseteq [n]} -U(b) = \min_{b \in \{0,1\}^n} \left( -p \circ x + \lambda \cdot p - \lambda \cdot (p \circ p) \right)^T b + \lambda \left( -M + p^T b \right)^2, \tag{16}$$

which matches the form of (13) with

$$a := -p \circ x + \lambda \cdot p - \lambda \cdot (p \circ p), \tag{17a}$$
$$\gamma := \lambda, \tag{17b}$$
$$\beta := -M, \tag{17c}$$
$$u := p. \tag{17d}$$

**Rounding the parameters.** We argue that rounding the parameters of an instance does not significantly affect the objective value. Consider rounded probabilities $\bar{p}$, with $|p_i - \bar{p}_i| \leq 1/D$ and rounded $a_i$ with $|a_i - \bar{a}_i| \leq 1/E$, for some integers $D$ and $E$ to be specified later. How much does the value of (13) change for the input (17) if these $u_i = p_i$ are changed to $\bar{u} := \bar{p}_i$ and $a_i$ to $\bar{a}_i$, regardless of $b$? For notational simplicity, we write the objective as

$$\Psi(b, a, \gamma, \beta, u) = \langle a, b \rangle + \gamma(\beta + \langle u, b \rangle)^2,$$

with the choice of variables specified in (17). Letting $p'_i := p_i - \bar{p}_i$ (for compactness), the difference before and after rounding is bounded by

$$\Delta = |\Psi(b, a, \gamma, \beta, u) - \Psi(b, \bar{a}, \gamma, \beta, \bar{u})|$$
$$\leq \left| (a - \bar{a})^T b \right| + \gamma \left| (p^T b)^2 - (\bar{p}^T b)^2 - 2M(p - \bar{p})^T b \right|$$
$$\leq \frac{n}{E} + \gamma \underbrace{\left( \left| (p^T b)^2 - (\bar{p}^T b)^2 \right| \right)}_{T} + \gamma \left( 2M \frac{n}{D} \right) \tag{18}$$

We bound the term $T$ by

$$
\begin{aligned}
T &= \sum_{i \neq j} b_i b_j (p_i p_j - \bar{p}_i \bar{p}_j) + \sum_i b_i^2 (p_i^2 - \bar{p}_i^2) + \\
&\leq \sum_{i \neq j} |p_i p_j - \bar{p}_i \bar{p}_j| + \sum_i |p_i^2 - \bar{p}_i^2| \\
&\leq \sum_{i \neq j} (\bar{p}_i p_j' + \bar{p}_j p_i' + p_i' p_j') + \sum_i (2\bar{p}_i p_i' + p_i'^2) \\
&\leq \sum_{i \neq j} (p_j' + p_i' + p_i' p_j') + \sum_i (2p_i' + p_i'^2) \\
&\leq 2n^2 \frac{1}{D} + n^2 \frac{1}{D^2} + n \frac{2}{D} + n \frac{1}{D^2} \\
&= \frac{n}{D} \left( 2(n+1) + \frac{n+1}{D} \right)
\end{aligned}
\tag{19}
$$

Plugging (19) back to (18), we have

$$
\Delta \leq \frac{n}{E} + \frac{\gamma n}{D} \left( 2(n+1) + \frac{n+1}{D} + 2M \right).
\tag{20}
$$

Recalling that $\gamma = \lambda$ for our problem. It can be verified by (20) that choosing $D \geq 2n\lambda(2M + 3(n+1))/\epsilon$ and $E \geq 2n/\epsilon$ ensures that $|\Psi(b, a, \gamma, \beta, u) - \Psi(b, a, \gamma, \beta, \bar{u})| \leq \epsilon$, for any arbitrary binary vector $b$.

We now define rounded versions of the problem parameters, which are rounded to increments of $D$. For all $i$, let

$$
\begin{aligned}
P_i &:= \lfloor D p_i \rfloor \\
\bar{u}_i &:= \frac{P_i}{D} = \frac{1}{D} \lfloor D p_i \rfloor \\
u_i' &:= P_i = \lfloor D p_i \rfloor,
\end{aligned}
$$

and $\beta' := -DM$ and $\gamma' := \frac{\gamma}{D^2} = \frac{\lambda}{D^2}$. Then letting $S(b)$ be the set indicated by $b$,

$$
\Psi(b, a, \gamma, \beta, u) = \langle a, b \rangle + \gamma(\beta + \langle u, b \rangle)^2 = -U(S(b))
$$

by (16), while

$$
\begin{aligned}
\Psi(b, \bar{a}, \gamma', \beta', u') &= \langle \bar{a}, b \rangle + \gamma'(\beta' + \langle u', b \rangle)^2 \tag{21} \\
&= \langle \bar{a}, b \rangle + \frac{\gamma}{D^2}(\beta D + \langle u', b \rangle)^2 \\
&= \langle \bar{a}, b \rangle + \gamma \left( \beta + \left\langle \frac{u'}{D}, b \right\rangle \right)^2 \\
&= \Psi(b, \bar{a}, \gamma, \beta, \bar{u}).
\end{aligned}
$$

We have just argued that $|U(S(b)) - \Psi(b, \bar{a}, \gamma, \beta, \bar{u})| \leq \epsilon$ when $D \geq 2n\lambda(2M + 3(n+1))/\epsilon$ and $E \geq 2n/\epsilon$. Observe also that the parameters $\bar{a}, \gamma', \beta', u'$ are such that (21) satisfies the requirements for Proposition 4. Therefore $\overline{OPT} \leftarrow -\text{R1UBQPSolver2}(\bar{a}, \gamma', \beta', u')$ is some objective value within $\pm\epsilon$ of our optimal value $U(S^*)$.

Algorithm 4 therefore begins by finding some value $\overline{OPT}$ which is the optimal value of the rounded instance of the problem realized by some $b \in \{0,1\}^n$ and within $\epsilon$ of $-U(S(b)f)$. Since whatever value the $b^*$ corresponding to $S^*$ attains on the rounded instance is within $\epsilon$ of $-U(S^*)$, it follows that this $b$ is an additive $\epsilon$-approximation to $U(S^*)$.

The remainder of $\text{ApproxL}_2$ is dedicated to reconstructing the set $S(b)$ itself. It does this by iteratively removing candidate components of the solution $i \in [n]$, determining whether or not each is necessesarily part of some such $S(b)$ (subject to the $i$ already discarded).

**Runtime.** R1UBQPSOLVER2 has runtime $O(K^4 E^4 n^5)$, and our reduction takes $K$ to be the maximum of $D$ and $\max(\lambda, x_{\max})/E$. In our reduction, $E = O((1 + \lambda)n(M + n)/\epsilon)$. Since APPROXL$_2$ makes one call to R1UBQPSOLVER and all other steps are negligible, it therefore runs in time $O(\frac{n^9(M+n)^4(1+\lambda)^4}{\epsilon^4})$.

In the case that $x_i \geq 0$ for all $i \in n$, we assume that $M \leq n$, since for $M \geq n$ it is optimal to take $S = [n]$; in this case the runtime guarantee is therefore $O(\frac{n^{13}(1+\lambda)^4}{\epsilon^4})$. $\qquad\square$

## B.6   Proof of Theorem 2

We follow the reduction outlined in Section 2 of Çela et al. [3]. In what follows, we reduce any instance of SUBSETSUM to an instance of our problem with the $L_2$ objective. An instance of SUBSETSUM is given by a set of positive integers $(t_1, \ldots, t_n)$ and a target sum $T$ that is also a positive integer. We assume without loss of generality that $t_i \leq T$ for each $i \in [n]$.

Given any instance of SUBSETSUM, we now construct an instance of our problem $\mathcal{I} = (x, p, M, \lambda)$ with the $L_2$ objective. Recall the notation $b \in \{0, 1\}^n$ with $b_i := \mathbb{1}\{i \in S\}$. Recall from (16) that our optimization problem can be written as:

$$\min_{b \in \{0,1\}^n} \left(-p \circ x + \lambda \cdot p - \lambda \cdot (p \circ p)\right)^T b + \lambda \left(-M + p^T b\right)^2,$$

where the first term is linear in $b$, and the second term is quadratic in $b$. We choose the problem parameters, such that the linear term becomes zero, and the quadratic term becomes the SUBSETSUM problem. For the linear term, we set $x_i := \lambda(1 - p_i)$ for each $i \in [n]$. It can be verified that $x_i \geq 0$, and that the linear term becomes zero. For the quadratic term, we set $M := 1$, and set the regularizer to be any value such that $\lambda > 0$. Then we set $p_i := \frac{t_i}{T}$, and we have $p_i \in [0, 1]$ by the assumption that $t_i \leq T$ for every $i \in [n]$. The optimization problem then becomes

$$\min_{b \in \{0,1\}^n} \lambda \left(-1 + \sum_{i \in [n]} b_i \frac{t_i}{T}\right)^2 = \min_{S \subseteq [n]} \frac{\lambda}{T^2} \left(-T + \sum_{i \in S} t_i\right)^2. \tag{22}$$

Note that the optimization problem (22) always attains a non-negative objective value, and attains an objective of zero if and only if $\sum_{i \in S} t_i = T$, that is, there exists a solution to the SUBSETSUM instance. Therefore, any algorithm for finding the optimal subset $S^*$ for our problem can be used to solve SUBSETSUM, completing the reduction. $\qquad\square$

## B.7   Proof of Theorem 3

We prove the three parts of the proposition separately. To prove upper bounds on the approximation ratio, we construct "bad" instances. Using the same argument as in the proof of Proposition 2 in Appendix B.4, the values $\{x_i\}_{i \in [n]}$ can be rescaled according to $\lambda$, and it suffices to construct instances for a fixed value of $\lambda > 0$.

To prove lower bounds for XGREEDY and XPGREEDY (across all instances), it is without loss of generality to assume that all $x_i \geq 0$. Recapp that since the loss $\rho = L_1^+$ is monotonic, adding an item never decreases the penalty term, and thus adding an item with negative value always decreases the overall objective. Moreover, all items with negative values appear at the end in the orders used by XGREEDY and XPGREEDY, so there are no more items with positive values once the two greedy algorithms reach the first negative item. Therefore, XGREEDY and XPGREEDY never choose solutions containing any item with negative value, and hence such items can be ignored for the purposes of these proofs.

### B.7.1   Proof of Theorem 3(a)

Let $M = 1$ and $\lambda > 1$. Consider an instance consisting of two items:

$$(x_1, p_1) = (0, 1)$$

$$(x_2, p_2) = \left(1, \frac{1}{2}\right).$$

It can be verified that PGREEDY only selects item 1, attaining an objective of 0. On the other hand, selecting item 2 attains a strictly positive objective of $\frac{1}{2}$.

### B.7.2 Proof of Theorem 3(b)

We separately prove the upper and lower bounds for XPGREEDY. We denote by $S_{XP}$ the selection found by XPGREEDY.

**Upper bound for XPGREEDY.** First, suppose that $p_{\min}$ is such that $1/p_{\min} \in \{2, 3, 4, \ldots\}$. We assume this without loss of generality in order to show that XPGREEDY is $O(p_{\min})$ for any $p_{\min} \in (0, 1]$. We may make this assumption because, first, for any $p_{\min} \in (1/2, 1]$ XPGREEDY must be an $O(1)$ approximation, and for $p_{\min}$ in this range it is therefore also $O(p_{\min})$. On the other hand, for any choice $p_{\min} \in (0, 1/2)$, consider the instance outlined below with $\frac{1}{\lceil 1/p_{\min} \rceil}$ as the minimum probability, together with an additional item for which $(p_i, x_i) = (p_{\min}, 0)$. Then XPGREEDY never chooses this last item, and the instance below demonstrates an upper bound of $O(\frac{1}{\lceil 1/p_{\min} \rceil}) = O(p_{\min})$.

We now construct the instance as follows. Let $M = 1$ and $\lambda = \frac{1+2c}{p_{\min}}$, and consider an unlimited number of items from two types:

$$(x_1, p_1) = \left( 1 + \frac{c}{2}, 1 \right)$$

$$(x_2, p_2) = \left( \frac{1}{p_{\min}}, p_{\min} \right).$$

We assume without loss of generality that $c \le 1/2$, since if $x_i \le (1 - c) \cdot \lambda$ is true for some $c > 1/2$ then it is true for $c \le 1/2$ also. Then it can be verified that $x_i \le (1 - c) \cdot \lambda$ for both types of items. Specifically

$$(1 - c) \cdot \lambda = \frac{(1 - c)(1 + 2c)}{p_{\min}} = \frac{1 + c - 2c^2}{p_{\min}} \ge 1 + c - 2c^2 \ge 1 + \frac{c}{2} = x_1$$

and

$$(1 - c) \cdot \lambda = \frac{1 + c - 2c^2}{p_{\min}} \ge \frac{1}{p_{\min}} = x_2$$

both hold for any $c \in (0, 1/4]$.

XPGREEDY adds items one-by-one from the first type. The objective after adding one item is $(1 + c/2)$. If a second item is added, the marginal change in the objective is $1 + \frac{c}{2} - \lambda = 1 + \frac{c}{2} - \frac{1+2c}{p_{\min}} < 0$. Hence, XPGREEDY selects exactly one item from the first type, attaining an objective of

$$U(S_{XP}) = 1 + c/2. \tag{23}$$

Now consider an alternative selection $S$ consisting of $t$ items of the second type. We now show that for some choice of $t$, the objective attained by the selection $S$ satisfies $U(S) \gtrsim \frac{1}{p_{\min}}$. We write the objective as

$$U(S) = R(S) - \lambda \cdot V(S)$$
$$= t - \lambda \cdot \mathbb{E}\left( |S_Z| - 1 \right)_+$$
$$= t - \lambda \left( \mathbb{E}|S_Z| - 1 + \mathbb{P}(|S_Z| = 0) \right)$$
$$= t - \lambda \left( t \cdot p_{\min} - 1 + (1 - p_{\min})^t \right).$$

Choosing $t = \frac{1}{p_{\min}}$, we have

$$U(S) = \frac{1}{p_{\min}} - \lambda \left( \frac{1}{p_{\min}} \cdot p_{\min} - 1 \right) - \lambda \left( 1 - p_{\min} \right)^{\frac{1}{p_{\min}}}$$
$$= \frac{1}{p_{\min}} - \lambda \left( 1 - p_{\min} \right)^{\frac{1}{p_{\min}}}$$
$$\ge \frac{1}{p_{\min}} - \frac{\lambda}{e}.$$

Substituting in $\lambda = \frac{1+2c}{p_{\min}}$, the objective is lower bounded by

$$U(S) \geq \frac{1}{p_{\min}} - \frac{1}{e} \cdot \frac{1+2c}{p_{\min}}$$

$$= \frac{1}{p_{\min}} \cdot \frac{e-1-2c}{e} \overset{(i)}{\geq} \frac{1}{p_{\min}}, \tag{24}$$

where step (i) is true by the assumption that $c \leq 1/4$.

Combining (23) and (24), we have an instance for which $U(S_{\mathrm{XP}}) \leq c' \cdot p_{\min} U(S)$ for some constant $c'$, establishing an upper bound of $O(p_{\min})$ on the approximation ratio.

**Lower bound for XPGREEDY.** First, if the total number of items is at most $M$, then it can be verified that selecting all items is optimal.

We denote by $S_M$ be the $M$ items with highest expected values $x_i p_i$, and denote by $S_{\mathrm{XP}}$ the solution that XPGREEDY finds. Moreover, we have $S_M \subseteq S_{\mathrm{XP}}$, because all values are assumed nonnegative, and the penalty term for the $L_1^+$ loss is zero when adding the first $M$ items. By definition, XPGREEDY only improves the objective in each step. Hence, we have

$$U(S_M) \leq U(S_{\mathrm{XP}}). \tag{25}$$

We now provide a lower bound for the selection $S_M$. Applying the Mean Bound (Lemma 2) with $\epsilon = \min(c, 1/2)$, we have either

$$|S^*| \leq \frac{c'}{p_{\min}} \tag{26a}$$

where $c'$ is a positive constant, or

$$\mu^* \leq \frac{101}{100} M. \tag{26b}$$

If (26a) holds, we have $|S^*| \leq \frac{cM}{p_{\min}}$ because $M \geq 1$. If (26b) holds, we have $p_{\min} \cdot |S^*| \leq \mu^* \leq \frac{101}{100} M$, and hence $|S^*| \leq \frac{101}{100} \cdot \frac{M}{p_{\min}}$. Combining the two cases, we have

$$|S^*| \lesssim \frac{M}{p_{\min}}. \tag{27}$$

Next, we consider the expected reward $R(S_M)$ for the selection $S_M$. We note that $|S^*| \geq |S_M| = M$ by the optimality of $S^*$. This is because $x_i \geq 0$, so adding any item to a selection containing less than $M$ items only increases the objective. Recall that the selection $S_M$ consists of the $M$ items with the maximum expected reward $p_i x_i$. Hence, the mean expected reward $p_i x_i$ for the set $S_M$ (over all items in this set) is greater than or equal to the mean expected reward for the set $S^*$. Namely,

$$\frac{1}{M} R(S_M) = \frac{1}{|S_M|} \sum_{i \in S_M} p_i x_i$$

$$\geq \frac{1}{|S^*|} \sum_{i \in S^*} p_i x_i = \frac{1}{|S^*|} R(S^*).$$

Hence, we have

$$U(S_M) = R(S_M) \geq \frac{M}{|S^*|} R(S^*)$$

$$\overset{(i)}{\gtrsim} p_{\min} \cdot R(S^*) \geq p_{\min} \cdot U(S^*), \tag{28}$$

where step (i) is true by plugging in (27). Combining (28) with (25), we have

$$U(S_{\mathrm{XP}}) \geq U(S_{\mathrm{XP}}) \gtrsim p_{\min} \cdot U(S^*),$$

completing the proof of the lower bound $\Omega(p_{\min})$ of the approximation ratio for XPGREEDY. $\qquad \square$

### B.7.3 Proof of Theorem 3(c)

We separately prove the upper and lower bounds for XGREEDY.

**Upper bound for XGREEDY.** Let $M = 1$ and $\lambda = \frac{2}{p}$. Consider an instance consisting of an unlimited number of items from two types:

$$(x_1, p_1) = (1, p_{\min})$$
$$(x_2, p_2) = (1 - \epsilon, 1),$$

where $\epsilon \in (0, 1)$ is a constant, and $p_{\min} \in (0, 1)$. We again suppose without loss of generality that $c \leq 1/2$, and it can be verified that $x_i \leq (1 - c) \cdot \lambda$ for both types of items.

XGREEDY adds items one-by-one from the first type. The objective after adding one item is $p_{\min}$. If a second item is added, the marginal change in the objective is $p_{\min} - \lambda p_{\min}^2 = -p_{\min} < 0$. Hence, XGREEDY selects exactly one item from the first type, attaining an objective of $p_{\min}$.

On the other hand, choosing a single item from the second type attains an objective value of $(1 - \epsilon)$. Therefore XGREEDY has a worst-case approximation ratio of at most $\frac{p_{\min}}{1-\epsilon}$, namely $O(p_{\min})$.

**Lower bound for XGREEDY.** We modify the construction of $S_M$ in the proof of part (b) to be the set of $M$ items with the highest values $x_i$, Then we apply similar arguments as in part (b), and outline the steps as follows.

Denote by $S_M$ the set of $M$ items with the highest values $x_i$, and denote by $S_X$ the solution that XGREEDY finds. Then again we have $S_M \subseteq S_X$ and hence

$$U(S_M) \leq U(S_X). \tag{29}$$

We now provide a lower bound for the selection $S_M$. Using the same argument as in part (b), we have (cf. (27)):

$$|S^*| \lesssim \frac{M}{p_{\min}}. \tag{30}$$

We note that $|S^*| \geq |S_M| = M$ by the optimality of $S^*$. Since the selection $S_M$ consists of the $M$ items with the maximum values $x_i$, the mean value for the set $S_M$ is greater than or equal to the mean value for the set $S^*$. Namely,

$$\frac{1}{M} \sum_{i \in S_M} x_i \geq \frac{1}{|S^*|} \sum_{i \in |S^*|} x_i.$$

Next note that for any $i \in S_M$, $\frac{1}{p_{\min}}(p_i x_i)$ is larger than $p_j x_j$ for any $j \in S^* \setminus S_M$, since for such $i$ and $j$ we have $\frac{1}{p_{\min}}(p_i x_i) \geq x_i \geq p_j x_j$. Therefore,

$$\begin{aligned}
U(S_M) = R(S_M) = \sum_{i \in S_M} p_i x_i &\geq p_{\min} \sum_{i \in S_M} x_i \\
&\geq p_{\min} \cdot \frac{M}{|S^*|} \sum_{i \in S^*} x_i \\
&\geq p_{\min} \cdot \frac{M}{|S^*|} \sum_{i \in S^*} p_i x_i \\
&\overset{(i)}{\gtrsim} p_{\min}^2 \cdot R(S^*) \geq p_{\min}^2 \cdot U(S^*), \tag{31}
\end{aligned}$$

where step (i) is true by plugging in (30). Combining (29) with (31) completes the proof of the lower bound $\Omega(p_{\min}^2)$ of the approximation ratio for XGREEDY. $\qquad\square$

### B.8 Proof of Theorem 4

**Notation.** We begin with some notation that is used in the proofs in this section. Given any instance $\{x_i, p_i\}_{i \in [n]}$, we construct a *rounded* instance $\{y_i, q_i\}_{i \in [n]}$ as follows. First we round *up* the

probabilities $p_i$ to $q_i := 2^{\lceil \log_2 p_i \rceil}$, that is, the smallest power of two that is greater than or equal to $p_i$. Then we construct new values $y_i$ such that the expected value of each item is preserved. Formally,

$$q_i := \min\left\{\frac{1}{2^i}, i \in \mathbb{N} : \frac{1}{2^i} \geq p_i\right\},$$

$$y_i := \frac{p_i}{q_i} x_i.$$

We slightly abuse the notation, and for any selection $S = \{x_i, p_i\}_{i \in [n]}$, we denote by $S' := \{y_i, q_i\}_{i \in [n]}$ the corresponding set with rounded probabilities and values. The parameters $M$ and $\lambda$ for this rounded instance remain unchanged. Note that by construction, we have

$$R(S) = R(S') \tag{32a}$$
$$V(S) \leq V(S') \tag{32b}$$
$$U(S) \geq U(S'), \tag{32c}$$

Eq. (32a) holds by the definition of the rounded set $S'$; Eq. (32b) in fact holds for all nondecreasing loss function $\rho$, because $\sum_{i \in S'} Z_i$ stochastically dominates $\sum_{i \in S} Z_i$; Eq. (32c) follows by combining (32a) and (32b).

Finally, recall the observation from Section 3.2.2 that we assume without loss of generality that $x_i > 0$ for all $i \in [S]$, since the marginal contribution of any $i$ for which $x_i \leq 0$ to any $U(S)$ is nonpositive.

**Overview of Algorithm 2.** We begin by reiterating the overview of Algorithm 2 presented in Section 3.2.2. At a high level, this algorithm proceeds first by dividing the items into three groups according to their values $x_i$.

$$N_{\mathrm{L}} := \{i \in [n] : x_i \leq (1 - \epsilon)\lambda\}$$
$$N_{\mathrm{M}} := \{i \in [n] : (1 - \epsilon)\lambda < x_i < \lambda\}$$
$$N_{\mathrm{H}} := \{i \in [n] : x_i \geq \lambda\}.$$

Since $U$ is submodular (see Lemma 1), the optimal solution within at least one of these groups is constant-competitive with $U(S^*)$. We consider each group separately, and obtain a constant-factor approximation for each group. We now provide an overview of the three cases. In particular, the small items in $N_{\mathrm{L}}$ are handled by Algorithm 5, and the medium items in set $N_{\mathrm{M}}$ are handled by Algorithm 6.

---

**Algorithm 5** LOWVALUEL$_1^+$ (with universal constant $c$)

---

**Require:** Problem instance $\mathcal{I} = (x, p, \lambda, M)$, with $x_i \leq (1 - \frac{p_{\min}}{4})\lambda$
**Ensure:** $S \subseteq [n]$ for which $U(S)/U(S^*) \gtrsim 1$
1: $\tau \leftarrow \frac{c}{p_{\min}^2} \max\left\{1, \log\left(\frac{1}{p_{\min}}\right), \log\left(\frac{\lambda}{x_{\max}}\right)\right\}$
2: $\mathcal{L} \leftarrow \{S \subseteq [n] : |S| \leq \tau\}$                           // Brute-force small instances
3: **for** $S \in \mathcal{L}$ **do**
4:      Calculate $U(S)$
5: $S_L \leftarrow \arg\max_{S \in \mathcal{L}} U(S)$
6: Let $q$ be the rounded $p$ and $Q \leftarrow \{q_i\}$ the distinct rounded probabilities; let $t_r$ be the multiplicity
        of each rounded probability $r$ in the vector $q$.                           // Round large instances
7: $\mathcal{H} \leftarrow \emptyset$
8: **for** $s \in \prod_{r \in Q}\{0, 1, \ldots, t_r\}$ **do**
9:      Construct $S$ from the $s_r$ many $i \in [n]$ of highest $x_i$ for which $q_i = r$, for each $r \in Q$
10:      Calculate $U(S)$, using unrounded probabilities and values
11:      $\mathcal{H} \leftarrow \mathcal{H} \cup \{S\}$
12: $S_H \leftarrow \arg\max_{S \in \mathcal{H}} U(S)$
13: **return** $S \in \{S_L, S_H\}$ maximizing $U(S)$

---

---

**Algorithm 6** MEDIUMVALUEL$_1^+$

---

**Require:** Problem instance $\mathcal{I} = (x, p, \lambda, M)$, with $(1 - \frac{p_{\min}}{4}) \cdot \lambda \leq x_i \leq \lambda$
**Ensure:** $S \subseteq [n]$ for which $U(S)/U(S^*) = \Omega(1)$
1: **if** $n \leq \frac{36}{p_{\min}^2}$ **then**
2:     **return** $\arg\max_{S \subseteq [n]} U(S)$
3: **else if** $\mu_{[n]} \geq M$ **then**
4:     Choose any $S \subseteq [n]$ such that $M \leq \mu_S < M + 1$
5: **else**
6:     Choose any $S \subseteq [n]$ such that $\frac{\mu_{[n]}}{3} \leq \mu_S \leq \frac{\mu_{[n]}}{2}$
7: **return** $S$

---

- **Low-value items $N_\mathbf{L}$ (Algorithm 5):** LOWVALUEL$_1^+$ presented in Algorithm 5 handles the case where items have low values. It consists of two parts: a search over small candidate solutions and a search over rounded candidate solutions. In the first part, we brute-force all small solutions whose size are at most $\tau$ (Line 3). This brute-force search succeeds if the optimal selection is small.

  The second part is the technical crux of proving the constant-factor approximation of LOWVALUEL$_1^+$. In the second part, we compute rounded probabilities and values $(q_i, y_i)$ for each item. This rounding procedure reduces the number of candidate solutions dramatically. We then brute-force over all rounded solutions (Line 8), select the rounded solution that maximizes the objective value, and prove that this solution is comparable to the (unrounded) optimal solution. Since the first part succeeds the case where the optimal solution is small, we may assume in this second part that the optimal solution is sufficiently large; this allows us to prove that our selection is robust to rounding.

  As an aside, we take the rounding to be to powers of two, but our analysis generalizes to rounding to powers of $(1 + c)$ for any constant $c > 0$, and this parameter may be tuned in order to trade off between runtime and performance in practice.

- **Medium-value items $N_\mathbf{M}$ (Algorithm 6):** MEDIUMVALUEL$_1^+$ presented in Algorithm 6 handles items with values close to $\lambda$. If the number of items is small, it brute-forces over all possible solutions (Line 2). If the number of items is large, the algorithm chooses any subset such that the expected number of accepted items is around $M$ (Line 4). If no such subset exists, then the expected number of accepted items when choosing all items must be less than $M$. In this case, then we choose a subset with approximately half the expected realizations compared to that of all items (Line 6). We choose a proportion less than one in order to ensure that the penalty incurred is not too large relative to the reward. This subset (Line 6) along with the subset defined in (Line 4) and always exists, formalized in the proof of Lemma 4 in Appendix B.8.2.

- **High-value items $N_\mathbf{H}$:** for the group of items with values above $\lambda$, it is easy to see that choosing the entire group is optimal.

We now prove the approximation ratio and runtime for ONESIDEDL$_1^+$.

**Proof of Theorem 4.** To begin, we split the items in the optimal set $S^*$, according to their values:

$$S_L^* := S^* \cap N_L,$$
$$S_M^* := S^* \cap N_M,$$
$$S_H^* := S^* \cap N_H.$$

By the submodularity of $U(S)$ in Lemma 1, we have $U(S_L^*) + U(S_M^*) + U(S_H^*) \geq U(S^*)$. In particular, this implies that $\max\{U(S_L^*), U(S_M^*), U(S_\mathrm{H}^*)\} \geq \frac{1}{3}U(S^*)$. In order to provide a constant-factor approximation to $U(S^*)$, it therefore suffices to identify sets which provide constant-factor approximations to $U(S_L^*)$, $U(S_M^*)$, and $U(S_\mathrm{H}^*)$, and return the set with the highest objective value among them. We choose $\epsilon := p_{\min}/4$ to determine the boundary between $N_L$ and $N_M$, and address each group separately. In each case we seek to find a subset which competes with the optimal subset of $N_L$ (say), which in turn is an approximation to $U(S_L^*)$.

**Low-value items $N_\mathrm{L}$.** The following lemma provides the approximation guarantee of LowValue$\mathrm{L}_1^+$.

**Lemma 3** (Small $x_i$). *Suppose that $x_i \leq \left(1 - \frac{p_{min}}{4}\right) \cdot \lambda$ for all $i \in [n]$. Then* LowValue$\mathrm{L}_1^+$ *(Algorithm 5) is a constant-factor approximation to $U(S^*)$ which runs in time $n^{\frac{c}{p_{min}^2} \max\left\{1, \log\left(\frac{1}{p_{min}}\right), \log\left(\frac{\lambda}{x_{max}}\right)\right\}}$, where $c$ is a universal constant.*

The proof of this lemma is provided in Appendix B.8.1, and is arguably the heart of the analysis of OneSided$\mathrm{L}_1^+$. By applying this lemma to $[n] = N_L$ we obtain $S_L$ with $U(S_L)$ within a constant factor to the optimal objective among selections within $N_\mathrm{L}$, and hence a constant factor to $U(S_L^*)$.

**Medium-value items $N_\mathrm{M}$.** The following lemma provides the approximation ratio guarantee of MediumValue$\mathrm{L}_1^+$.

**Lemma 4** (Medium $x_i$). *If $\lambda\left(1 - \frac{p_{min}}{4}\right) \leq x_i \leq \lambda$ for all $i \in [n]$ then* MediumValue$\mathrm{L}_1^+$ *(Algorithm 6) finds some $S \subseteq [n]$ which is a constant-factor approximation to $U(S^*)$ and runs in time $n^{O(1/p_{min}^2)}$.*

The proof of this lemma is provided in Appendix B.8.2. By applying this lemma to $[n] = N_M$ we obtain $S_M$ with $U(S_M)$ within a constant factor to the optimal objective among all selections within $N_\mathrm{M}$, and hence a constant factor to $U(S_M^*)$.

**High-value items $N_\mathrm{H}$.** This case is simple: we select all items by taking $S_H = N_H$. It can be verified that adding every item strictly increases the objective, and hence $N_\mathrm{H}$ attains the optimal objective among all selections within $N_\mathrm{H}$. By the optimality of $S_H$, we have $U(S_H) = U(S_H^*)$.

Putting the three cases together, we have $S_L$, $S_M$, and $S_H$, and by the argument provided above at least one of these is a constant-factor approximation to $U(S^*)$. Therefore choosing the one with highest objective value gives a constant-factor approximation.

**Runtime.** The algorithms for the cases above operate by identifying a collection of sets to test the objective value of, and then evaluating the objective. Fortunately this can be done efficiently.

**Lemma 5** (Efficient Objective Evaluation). *Suppose that $\rho(a, M)$ is known for all $a \in \{0, 1, \ldots, n\}$. For any set of items $\{x_i, p_i\}_{i\in[n]}$, the objective $U([n])$ can be computed in $O(n^2)$ arithmetic operations.*

This is proved in Appendix B.8.3. Applying this lemma to any candidate subset $S$ shows that the objective with respect $S \subseteq [n]$ can be computed in $O(|S|^2)$ arithmetic operations.

By Lemma 3, the runtime of LowValue$\mathrm{L}_1^+$ is $n^{\frac{c}{p_{\min}^2} \max\left\{1, \log\left(\frac{1}{p_{\min}}\right), \log\left(\frac{\lambda}{x_{\max}}\right)\right\}}$. By Lemma 4 the runtime of MediumValue$\mathrm{L}_1^+$ is $n^{O(1/p_{\min}^2)}$, which is less than that of LowValue$\mathrm{L}_1^+$. Finally, the high-value items case entails evaluating the objective in of a single set; by Lemma 5 this can be done in $O(n^2)$.

The cost of combining these cases is polynomial in $n$, and so the brute force stage of LowValue$\mathrm{L}_1^+$ dictates the runtime of OneSided$\mathrm{L}_1^+$, giving the claimed runtime of $n^{c\frac{1}{p_{\min}^2} \max\left\{1, \log\left(\frac{1}{p_{\min}}\right), \log\left(\frac{\lambda}{x_{\max}}\right)\right\}}$ for some universal constant $c > 0$. $\qquad\square$

We now turn to the statements and proofs of the supporting lemmas.

The following lemma says that the solution can be downsampled so that its $\sum_i p_i$ is at most a constant factor from the original, while $\sum_i p_i x_i$ is at least a constant factor from the original. Informally, we simply select the appropriate number of items with the highest $x_i$.

**Lemma 6** (Downsampling Lemma). *Consider an instance $\{p_i, x_i\}_{i\in[n]}$ with $x_i \geq 0$ for all $i \in [n]$. Then for any $S \subseteq [n]$ and any $\beta \in [0, 1]$, there exists some $T \subseteq S$ that satisfies*

$$\mu_T \leq \beta \cdot \mu_S \tag{33a}$$

*and*

$$R(T) \geq \beta \left(1 - \frac{1}{\beta \cdot p_{min} \cdot |S|}\right) \cdot R(S). \tag{33b}$$

The proof of this lemma is provided in Appendix B.8.4. If we were allowed to choose items to be in $T$ fractionally, then condition (33b) would more closely mimic condition (33a) and the proof of this lemma would be even more straightforward; as it is, condition (33b) must be slightly weaker since we must sometimes leave the last item out of $T$ in order to satisfy condition (33a).

This lemma supports the efficient search over rounded solutions which is conducted in $\textsc{LowValueL}_1^+$. Informally, it does this by proving that if some starting set satisfies certain properties, then either there is a small subset with good objective value, or the search over rounded solutions will identify a subset with good objective value.

**Lemma 7** (Rounding Lemma). *Consider the one-sided $\rho = L_1^+$ loss. Consider any selection $S \subseteq [n]$ that simultaneously satisfies*

$$U(S) \geq 0 \tag{34a}$$

$$\mu_S \leq \frac{3}{2}M \tag{34b}$$

$$\lambda V(S) \leq \frac{1}{15}R(S). \tag{34c}$$

*Then there exists some subset $T \subseteq S$ that satisfies either*

$$U(T) \geq \frac{1}{3}U(S) \qquad and \qquad |T| \leq \frac{24}{p_{min}}, \tag{35a}$$

*or*

$$U(T) \geq U(T') \geq \frac{1}{24}U(S), \tag{35b}$$

*where $T'$ denotes the rounded instance of the set $T$.*

The proof of this lemma is provided in Appendix B.8.5.

This next lemma bounds the penalty of a subset with expected realized size smaller than $M$. It uses the independence of the events $Z_i$ to apply tail bounds to the probability that the realized size of the subset exceeds $M$. When applied to a downsampled subset derived from Lemma 6, it will show that the penalty decreases exponentially while the reward decreases only linearly, yielding a subset which is within a small factor of the starting set's objective but is much less balanced.

**Lemma 8** (Downsampling Penalty Bound). *Consider the one-sided $\rho = L_1^+$ loss. Consider any selection $S \subseteq [n]$ such that $\mu_S \leq M$. Then for all $k \in \mathbb{N}_+$, the penalty term is bounded as*

$$V(S) \leq \lambda \cdot k \cdot e^{\frac{-2(M-\mu_S)^2}{|S|}} \cdot \left(1 - e^{\frac{-4(M-\mu_S)k}{|S|}}\right)^{-2}. \tag{36}$$

The proof of this lemma is provided in Appendix B.8.6. Informally, under this loss the penalty increases linearly in the extent to which the realized size of $S$ exceeds $M$, while the probability that such a violation occurs decreases exponentially. The parameter $k$ is the size of the buckets for which the analysis of these competing influences is performed.

### B.8.1 Proof of Lemma 3

Recall that we define $\tau := \frac{c}{p_{\min}^2} \max\left\{1, \log\left(\frac{1}{p_{\min}}\right), \log\left(\frac{\lambda}{x_{\max}}\right)\right\}$ in Line 2 of Algorithm 5. Let $c_0$ be the universal constant identified in Lemma 2. With $p_{\min} \leq 1$, it is straightforward to verify that there exists a universal constant $c$, such that $\tau$ is bounded by

$$\tau > \frac{c_0}{p_{\min}} \log\left(\frac{4}{p_{\min}}\right), \tag{37a}$$

$$\tau \geq \frac{24}{p_{\min}} \tag{37b}$$

$$\tau \geq \frac{9}{2}\left(\frac{1}{p_{\min}^2}\left(7 + \log\left(\frac{1}{p_{\min}}\right) + 3\log\left(\frac{\lambda}{x_{\max}}\right)\right)\right) \tag{37c}$$

We use these bounds in the remaining proof.

In Line 2-5, we evaluate the objective for each selection $S$ with $|S| \leq \tau$ by brute-force. Hence, if $|S^*| \leq \tau$, then the optimal selection is correctly identified. It remains to consider the case when $|S^*| > \tau$.

When $|S^*| > \tau$, we apply the Mean Bound (Lemma 2) with $\epsilon = \frac{p_{\min}}{4}$. We have either

$$|S^*| \leq \frac{c_0}{p_{\min}} \log \left( \frac{4}{p_{\min}} \right) \tag{38a}$$

or

$$\mu^* \leq \frac{101}{100} M. \tag{38b}$$

The bound (37a) on $\tau$ contradicts (38a). Hence we have that (38b) holds, namely $\mu^* \leq \frac{101}{100} M$ from.

We call a set $S$ "balanced" if $\lambda V(S) > \frac{1}{15} R(S)$, that is, the penalty term is a nontrivial portion of the reward term. Otherwise, we call the set "unbalanced". We consider the following two cases separately depending on whether the set $S^*$ is balanced or not.

**Case 1:** $|S^*| > \tau$ **and** $\lambda V(S^*) \leq \frac{1}{15} R(S^*)$**.**

Note that the optimal selection always has a nonnegative objective for the $L_1^+$ loss. That is, $U(S^*) \geq 0$. Hence, conditions (34) are satisfied. Applying the Rounding Lemma (Lemma 7), there exists some $T \subseteq S^*$ such that

$$U(T) \geq \frac{1}{3} U(S^*) \qquad \text{and} \qquad |T| \leq \frac{24}{p_{\min}}, \tag{39a}$$

or

$$U(T) \overset{(i)}{\geq} U(T') \geq \frac{1}{24} U(S^*), \tag{39b}$$

In this first case (39a), by the bound (37b) on $\tau$, we have

$$\tau \geq \frac{24}{p_{\min}} \geq |T|.$$

Hence, the selection $T$ is included in the brute-force search. We obtain a constant-factor approximation to $U(S^*)$ in the brute-force search over small solutions (Line 5 of Algorithm 5).

In the second case (39b), if $|T| \leq \tau$, then again the selection $T$ is included in the brute-force search in Line 5 of Algorithm 5, and the brute-force identifies a solution which is at least as good and hence a constant-factor approximation. If $|T| > \tau$, then Line 6 to Line 12 search over all possible rounded solutions, including $T'$ which is a constant-factor approximation due to (39b). Hence, it identifies a solution which is at least as good and hence a constant-factor approximation. identifies some $\widehat{T}$ for which $U(\widehat{T}) \geq U(\widehat{T}') \geq U(T')$, which provides a constant-factor approximation to $U(S^*)$.

**Case 2:** $|S^*| > \tau$ **and** $\lambda V(S^*) > \frac{1}{15} R(S^*)$**.**

As an overview of this case, we appeal to the Downsampling Lemma (Lemma 6) with a small downsampling ratio in order to obtain some $T \subseteq S$, and then argue that $\lambda V(T) \leq \frac{1}{15} R(T)$. Then we obtain a constant-factor approximation to $U(T)$ by solving the rounded problem in Case 1.

**Downsampling to an unbalanced set.** Starting with the optimal selection $S^*$, we apply the Downsampling Lemma (Lemma 6) construction some $T \subseteq S^*$ such that this $T$ is unbalanced but still yields a large objective. Specifically, applying Lemma 6 with $\beta = \frac{1}{2}$, there exists some $T \subseteq S^*$ that satisfies $\mu_T \leq \frac{\mu^*}{2}$ and

$$R(T) \geq \left( \frac{1}{2} - \frac{1}{|S^*| \cdot p_{\min}} \right) R(S^*). \tag{40}$$

We assume that the selection $T$ is sufficiently unbalanced as:

$$V(T) \leq \frac{R(T)}{15}. \tag{41}$$

We now identify a constant-factor approximation by similar arguments as in Case 1. Specifically, under the assumption (41), the objective of $T$ satisfies

$$U(T) \overset{(i)}{\geq} \frac{14}{15} R(T) \overset{(ii)}{\geq} \frac{14}{15} \cdot \left( \frac{1}{2} - \frac{1}{|S^*| \cdot p_{\min}} \right) R(S^*)$$
$$\geq \frac{14}{15} \cdot \left( \frac{1}{2} - \frac{1}{|S^*| \cdot p_{\min}} \right) U(S^*), \tag{42}$$

where step (i) is due to the assumption (41), and step (ii) is due to (40). By the assumption $|S^*| \geq \tau$ and the bound (37b) on $\tau$, we have

$$|S^*| \geq \tau \geq \frac{24}{p_{\min}}. \tag{43}$$

Applying (43) to inequality (42), the selection $T$ is a constant-factor approximation to $S^*$ with $U(T) \geq \frac{1}{4} U(S^*)$. Since $T$ is sufficiently unbalanced by assumption (41), using the same arguments as in Case 1 to the set $T$ identifies a selection that is a constant-factor approximation to $T$, and therefore to $U(S^*)$. It now remains to prove (41).

**Proving (41).** Recall from (38b) that $\mu^* \leq \frac{101}{100} M$. Hence, we have $\mu_T \leq \frac{\mu^*}{2} \leq \frac{101}{200} M < M$. Then we provide an upper bound on the variance term $V(T)$ by applying Lemma 8 to the set $T$. Applying Lemma 8 with $k = 1$, we have

$$V(T) \leq \lambda \cdot \underbrace{\text{Exp}\left( \frac{-2(M - \mu_T)^2}{|T|} \right)}_{T_1} \cdot \underbrace{\left( 1 - e^{\frac{-4(M - \mu_T)}{|T|}} \right)^{-2}}_{T_2}. \tag{44}$$

We bound the two terms in (44) separately.

Recall from (38b) that $\mu^* \leq \frac{101}{100} M$. We then have

$$M - \mu_T \geq \left( \frac{100}{101} - \frac{1}{2} \right) \mu^* > \frac{1}{3} \mu^*. \tag{45}$$

Using (45), we bound the term $T_1$ as

$$T_1 = \text{Exp}\left( \frac{-2(M - \mu_T)^2}{|T|} \right) \leq \text{Exp}\left( -\frac{2(\mu^*)^2}{9|T|} \right) \overset{(i)}{\leq} \text{Exp}\left( -\frac{2}{9} p_{\min}^2 \cdot |S^*| \right)$$
$$\overset{(ii)}{\leq} \frac{1}{720} \frac{p_{\min}^3 x_{\max}}{\lambda}$$

where step (i) is true by plugging in $\mu^* \geq |S^*| \cdot p_{\min}$, and $|S^*| \geq |T|$ due to $T \subseteq S^*$; step (ii) is true by the fact that $|S^*| \geq \tau$ with (37c). Let $i_{max}$ be the item with the highest value. With $M \geq 1$, the utility for selecting the item with the highest value is $U(\{i_{\max}\}) = R(\{i_{\max}\}) = p_{i_{\max}} x_{\max} \geq p_{\min} x_{\max}$. Hence, the reward of the optimal selection is bounded by $R(S^*) \geq U(S^*) \geq U(\{i_{\max}\}) \geq p_{\min} x_{\max}$. Hence,

$$T_1 \leq \frac{p_{\min}^2}{45} \frac{R(S^*)}{\lambda}, \tag{46}$$

Using again $M - \mu_T \geq \frac{1}{3} \mu^* \geq \frac{1}{3} \cdot |S^*| \cdot p_{\min}$ and $|S^*| \geq |T|$, we bound the term $T_2$ as

$$T_2 = \left( 1 - e^{\frac{-4(M - \mu_T)}{|T|}} \right)^{-2} \leq \left( 1 - e^{\frac{-4\mu^*}{9|S^*|}} \right)^{-2} \leq (1 - e^{-\frac{4}{9} p_{\min}})^{-2} \leq \frac{16}{p_{\min}^2}, \tag{47}$$

where step (i) is true because it can be shown by algebra that

$$1 - e^{-\frac{4}{9} p} - \frac{1}{4} p \geq 0 \qquad \text{for every } p \in [0, 1].$$

Plugging term $T_1$ from (46) and term $T_2$ from (47) back to (44) yields

$$V(T) \leq \lambda \cdot \frac{1}{45} R(S^*) \overset{(i)}{\leq} \frac{1}{15\lambda} \left( \frac{1}{2} - \frac{1}{|S^*| \cdot p_{\min}} \right) R(S^*) \overset{(ii)}{\leq} \frac{R(T)}{15\lambda}, \tag{48}$$

where step (i) is true by $|S^*| \geq \tau \geq \frac{24}{p_{\min}}$ due to (37b), and step (ii) is true due to (40), proving (41).

**Runtime.** We conclude by analyzing the runtime of $\textsc{LowValueL}_1^+$.

The number of sets $S$ such that $|S| \le \tau$ is bounded by $|\mathcal{L}| = 2^\tau \le n^\tau$. For each $S \in \mathcal{L}$, we compute $U(S)$ in $O(\tau^2)$. By Lemma 5, the objective of each such set may be evaluated in $O(\tau^2)$ operations, and so the runtime of evaluating the objective for all of these small subsets is $n^{O(\tau)}$.

We also compute the objective for the $O(n^{|Q|})$ rounded sets identified in Algorithm 5, where $|Q| \le \left\lceil \log_2(\frac{1}{p_{\min}}) \right\rceil$, which again by Lemma 5 can be done in $O(n^2)$ operations per set. This is therefore $n^{O(\tau)}$ also.

All of the other simple steps of $\textsc{LowValueL}_1^+$ are also polynomial in $n$ or $\tau$. Therefore, its overall runtime is $n^{O(\tau)}$. $\qquad\square$

### B.8.2 Proof of Lemma 4

First, we observe that when $n \le \frac{36}{p_{\min}^2}$, we find the optimal solution exactly by brute forcing over all possible solutions $S \subseteq [n]$ (Line 2 of Algorithm 6). Hence, in the rest of the proof we assume that $n \ge \frac{36}{p_{\min}^2}$. We discuss the two cases of $\mu_{[n]} \le M$ (Line 6) and $\mu_{[n]} > M$ (Line 4) separately.

We start by establishing a reformulation of the objective under $L_1^+$ penalty which is convenient when all items have value close to $\lambda$, and a pair of upper and lower bounds.

**Bounding the objective.** Recall that our objective is of the form $U(S) = \mathbb{E}_Z[F_S(Z)]$, where the random vector $Z \in \{0,1\}^n$ is the Bernoulli realization of each item, and $F_S(Z)$ is the realized utility:

$$
\begin{aligned}
F_S(Z) &:= \sum_{i \in S} Z_i x_i - \lambda \left( \sum_{i \in S} Z_i - M \right)_+ \\
&= \sum_{i \in S} Z_i x_i - \lambda \cdot \max\left( 0, \ \sum_{i \in S} Z_i - M \right) \\
&= \min\left( \sum_{i \in S} Z_i x_i, \ \sum_{i \in S} Z_i x_i - \lambda \cdot \left( \sum_{i \in S} Z_i - M \right) \right). \qquad (49)
\end{aligned}
$$

Plugging in the assumption that $(1-\epsilon) \cdot \lambda \le x_i \le \lambda$ to (49), and recalling that $|S_Z| := \sum_{i \in S} Z_i$, we derive the upper bound

$$
F_S(Z) \le \min\left\{ \lambda \cdot |S_Z|, \lambda \cdot M + \sum_{i \in S} Z_i(x_i - \lambda) \right\}
$$
$$
\frac{F_S(Z)}{\lambda} \le \min\left\{ |S_Z|, \ M \right\}, \qquad (50a)
$$

and the lower bound

$$
F_S(Z) \ge \min\left\{ (1-\epsilon)\lambda \cdot |S_Z|, (1-\epsilon)\lambda \cdot |S_Z| - \lambda(|S_Z| - M) \right\}
$$
$$
\frac{F_S(Z)}{\lambda} \ge \min\left\{ (1-\epsilon) \cdot |S_Z|, \ M - \epsilon \cdot |S_Z| \right\}. \qquad (50b)
$$

We denote $\epsilon = \frac{p_{\min}}{4}$ for notational simplicity. As an overview, for the two cases to be presented below, we apply the upper bound (50a) to the optimal set $S^*$, and the lower bound (50b) to our candidate sets which we are proving are competitive with $S^*$.

**Case 1: $\mu_{[n]} > M$ and $n \ge \frac{36}{p_{\min}^2}$.** Consider any arbitrary set $S \subseteq [n]$ that satisfies (cf. Line 4 of Algorithm 6):

$$
M \le \sum_{i \in S} p_i < M + 1. \qquad (51)
$$

Such a set $S$ always exists because each $p_i \leq 1$. Furthermore, such a set in $S$ can be found efficiently, by greedily adding items to the set in an arbitrary order one-by-one until condition (51) is satisfied. We denote by $\mathcal{E}$ the event that at most half of the Bernoulli random variables from $S$ are 1. Formally, $\mathcal{E} := \{|S_Z| \leq M/2\}$. By the multiplicative Chernoff bound,

$$\mathbb{P}(\mathcal{E}) = \mathbb{P}\left(|S_Z| \leq \frac{M}{2}\right) \overset{(i)}{\leq} \Pr\left(|S_Z| \leq \frac{\mu_S}{2}\right) \leq e^{-\frac{\mu_S}{8}} \overset{(ii)}{\leq} e^{-\frac{M}{8}}, \tag{52}$$

where steps (i) and (ii) are true by the construction of $S$ in (51). We derive a lower bound on $F_S(Z)$ depending on $\mathcal{E}$. Conditional on $\mathcal{E}$, the penalty term is 0 and we have $F_S(Z) \geq 0$. We now consider the case conditional on $\overline{\mathcal{E}}$. By the assumption of $\mu_S < M + 1$ from (51), we have the deterministic relation $|S_Z| \leq \frac{M+1}{p_{\min}}$. Applying the lower bound in (50b), conditional on $\overline{\mathcal{E}}$,

$$F_S(Z) \geq \lambda \cdot \min\left\{(1 - \epsilon) \cdot \frac{M}{2}, \ M - \epsilon \cdot |S_Z|\right\}$$

$$\geq \lambda \cdot \min\left\{\frac{(1 - \epsilon)M}{2}, \ M - \frac{\epsilon(M + 1)}{p_{\min}}\right\}$$

$$= \lambda M \cdot \min\left\{\frac{1 - \epsilon}{2}, \ 1 - \frac{\epsilon(M + 1)}{M \cdot p_{\min}}\right\}$$

$$\overset{(i)}{=} \lambda M \cdot \frac{1 - \epsilon}{2}$$

where step (i) holds because by the assumption $\epsilon \leq \frac{p_{\min}}{4}$ and $M \geq 1$ (recall that $M \in \mathbb{N}_+$), we have $\frac{\epsilon}{p_{\min}} \leq \frac{1}{4}$ and $\frac{M+1}{M} \leq 2$. Therefore, we have $1 - \frac{\epsilon(M+1)}{M \cdot p_{\min}} \geq \frac{1}{2}$. Taking an expectation over $Z$, we then have

$$U(S) = \mathbb{E}[F_S(Z)]$$

$$\geq 0 \cdot \Pr(\mathcal{E}) + \frac{(1 - \epsilon)\lambda M}{2} \cdot \Pr(\overline{\mathcal{E}})$$

$$\overset{(i)}{\geq} \frac{(1 - \epsilon)(1 - e^{-\frac{M}{8}})}{2} \cdot \lambda M$$

$$\overset{(ii)}{\geq} \frac{(1 - \epsilon)(1 - e^{-\frac{M}{8}})}{2} \cdot U(S^*),$$

where step (i) is true by (52), and step (ii) is true by applying (50a) to the optimal selection $S^*$. Since $\epsilon = \frac{p_{\min}}{4} \leq \frac{1}{4}$ and $M \geq 1$ by assumption, this guarantees a constant-factor approximation of $S$ to the optimal subset $S^*$.

**Case 2: $\mu_{[n]} \leq M$ and $n \geq \frac{36}{p_{\min}^2}$.** In this case $n$ is large enough to apply concentration bounds, so we downsample the set $[n]$ by a factor of two and discount the probability that $|S_Z|$ exceeds $M$. This differs from Case 1 in that we cannot compare our objective against $\lambda \cdot M$. In particular, we consider any arbitrary set $S \subseteq [n]$ that satisfies (cf. Line 6 of Algorithm 6):

$$\frac{\mu_{[n]}}{3} \leq \mu_S \leq \frac{\mu_{[n]}}{2} \leq \frac{M}{2}. \tag{53}$$

We first show that such a set $S$ always exists. Note that we have $\mu_{[n]} \geq np_{\min} \geq np_{\min}^2 \geq 36$ by the assumption of $n \geq \frac{36}{p_{\min}^2}$. Hence, $\frac{\mu_{[n]}}{2} - \frac{\mu_{[n]}}{3} \geq 1$. Since each $p_i \leq 1$, a set $S$ always exists. Moreover, it can be found efficiently, by greedily adding items one-by-one in any arbitrary order until condition (53) is satisfied. In what follows, we separately bound the values of $U(S)$ and $U(S^*)$. We use an intermediate quantity of the expectation of the random variable $|S_Z|$ truncated at $2\mu_S$, defined by

$$G(|S_Z|) := \mathbb{E}\left[|S_Z| \cdot \mathbb{1}\{|S_Z| \leq 2\mu_S\}\right].$$

**Lower bound on $U(S)$.** Due to the condition (53) that $\mu_S \leq \frac{M}{2}$, we have

$$G(|S_Z|) := \mathbb{E}\left[|S_Z| \cdot \mathbb{1}\{|S_Z| \leq 2\mu_S\}\right]$$

$$\leq \mathbb{E}\left[|S_Z| \cdot \mathbb{1}\{|S_Z| \leq M\}\right]. \tag{54}$$

We claim the deterministic relation

$$\min\left\{|S_Z|, \frac{M - \epsilon|S_Z|}{(1-\epsilon)}\right\} \geq |S_Z| \cdot \mathbb{1}\{|S_Z| \leq M\}. \tag{55}$$

To see (55), we observe that when $|S_Z| \leq M$, the left-hand side has

$$\frac{1}{1-\epsilon}(M - \epsilon|S_Z|) \geq M \geq |S_Z|. \tag{56a}$$

When $|S_Z| > M$, the right-hand side is zero, and the left-hand side is nonnegative, because

$$M \geq 2\mu_S \geq 2p_{\min}|S| \geq 2p_{\min}|S_Z| \geq \epsilon|S_Z|. \tag{56b}$$

Plugging (55) to (54), we have

$$\begin{aligned}
G(|S_Z|) &\leq \mathbb{E}\min\left\{|S_Z|, \frac{M - \epsilon|S_Z|}{1-\epsilon}\right\} \\
&\leq \frac{1}{1-\epsilon}\mathbb{E}\min\left\{|S_Z|, \frac{M - \epsilon|S_Z|}{1-\epsilon}\right\} \\
&= \frac{1}{(1-\epsilon)^2}\mathbb{E}\left[\min\{(1-\epsilon)\cdot|S_Z|, M - \epsilon\cdot|S_Z|\}\right] \\
&\overset{(i)}{\leq} \frac{1}{(1-\epsilon)^2}\cdot\frac{U(S)}{\lambda},
\end{aligned} \tag{57}$$

where step (i) follows from (50b).

**Upper bound on $U(S^*)$.**  We decompose the expectation of $|S_Z|$ as

$$\mathbb{E}|S_Z| = G(|S_Z|) + \mathbb{E}\left[|S_Z|\cdot\mathbb{1}\{|S_Z| > 2\mu_S\}\right],$$

and hence

$$\begin{aligned}
G(|S_Z|) &= \mu_S - \mathbb{E}\left[|S_Z|\cdot\mathbb{1}\{|S_Z| > 2\mu_S\}\right] \\
&\overset{(i)}{\geq} \mu_S - \mathbb{E}\left[|S_Z|\cdot\mathbb{1}\{|S_Z| > M\}\right] \\
&= \mu_S - \mathbb{E}\left[M\cdot\mathbb{1}\{|S_Z| > M\} + (|S_Z| - M)\cdot\mathbb{1}\{|S_Z| > M\}\right] \\
&= \mu_S - \underbrace{M\cdot\mathbb{P}\Big(|S_Z| > M\Big)}_{T_1} - \underbrace{\mathbb{E}\left[(|S_Z| - M)_+\right]}_{T_2},
\end{aligned} \tag{58}$$

where step (i) is true by the condition (53) that $\mu_S \leq \frac{M}{2}$. We now analyze the two terms $T_1$ and $T_2$ separately. We define $\delta$ such that $M = (1+\delta)\mu_S$, and we have $\delta \geq 1$ by the construction of $S$ in (53).

For the term $T_1$, we apply the multiplicative Chernoff bound. We have

$$\mathbb{P}\Big(|S_Z| > M\Big) \leq \mathbb{P}\Big(|S_Z| > 2\mu_S\Big) \leq e^{-\frac{\mu_S}{3}}.$$

Then

$$T_1 \leq 2\mu_S\cdot e^{-\frac{\mu_S}{3}} \overset{(i)}{\leq} \frac{e}{6}, \tag{59}$$

where it can verified that step (i) holds for any $\mu_S \in \mathbb{R}$.

For the term $T_2$, note that $T_2 = \frac{V(S)}{\lambda}$, and by condition (53) we have $\mu_S \leq \frac{M}{2} \leq M$. Applying Lemma 8 with $k = \lceil\frac{1}{p_{\min}}\rceil$ yields

$$T_2 \leq \left\lceil\frac{1}{p_{\min}}\right\rceil\cdot\text{Exp}\left(\frac{-2(M-\mu_S)^2}{|S|}\right)\cdot\left(1 - e^{\frac{-4(M-\mu_S)}{|S|}\lceil\frac{1}{p_{\min}}\rceil}\right)^{-2}.$$

Using the relations $|S| \leq \frac{\mu_S}{p_{\min}}$ and $M - \mu_S \geq \mu_S$, we have

$$T_2 \leq \left\lceil \frac{1}{p_{\min}} \right\rceil \cdot \mathrm{Exp}\left( \frac{-2\mu_S^2}{\mu_S/p_{\min}} \right) \cdot \left( 1 - e^{\frac{-4\mu_S}{\mu_S/p_{\min}} \left\lceil \frac{1}{p_{\min}} \right\rceil} \right)^{-2}$$

$$= \left\lceil \frac{1}{p_{\min}} \right\rceil \cdot \mathrm{Exp}\left( -2p_{\min}\mu_S \right) \cdot \left( 1 - e^{-4} \right)^{-2}$$

$$\leq \left\lceil \frac{1}{p_{\min}} \right\rceil \cdot \left( 1 - e^{-4} \right)^{-2}. \tag{60}$$

Plugging term $T_1$ from (59) and term $T_2$ from (60) back to (58) yields

$$G(|S_Z|) \geq \mu_S - \frac{6}{e} - 1.04 \cdot \left\lceil \frac{1}{p_{\min}} \right\rceil.$$

Recall from the construction of $S$ in (53) that $\frac{\mu_{[n]}}{3} \leq \mu_S \leq \frac{\mu_{[n]}}{2}$. Furthermore, by the assumption that $n \geq \frac{36}{p_{\min}^2}$, we have

$$\mu_{[n]} \geq np_{\min} \geq \frac{36}{p_{\min}} \geq \max\left\{ 36, 18 \left\lceil \frac{1}{p_{\min}} \right\rceil \right\}. \tag{61}$$

Hence, we have

$$G(|S_Z|) \geq \frac{\mu_{[n]}}{3} - \frac{\mu_{[n]}}{12} - \frac{\mu_{[\mu_S]}}{12} \geq \frac{\mu_{[n]}}{6}.$$

Applying inequality (50a) with the fact that $\mathbb{E}[|S_Z|] \leq \mu_{[n]}$, we have

$$U(S^*) \leq \lambda \cdot \mu_{[n]}$$

and hence

$$G(|S_Z|) \geq \frac{\mu_{[n]}}{6} \geq \frac{U(S^*)}{6\lambda}. \tag{62}$$

Finally, combining (57) and (62) yields

$$\frac{U(S)}{U(S^*)} \geq \frac{\lambda(1-\epsilon)^2 \cdot G(|S_Z|)}{6\lambda \cdot G(|S_Z|)} = \frac{(1-\epsilon)^2}{6},$$

yielding a constant-factor approximation with $\epsilon = \frac{p_{\min}}{4} \leq \frac{1}{4}$.

**Runtime.** MEDIUMVALUEL$_1^+$ (Algorithm 6) begins by brute forcing over small sets, and there are $2^n \leq 2^{36/p_{\min}^2} \leq n^{36/p_{\min}^2}$ such sets. By Lemma 5, the objective value for each such set can be evaluated in polynomial time, and so the runtime in this case is $n^{3+36/p_{\min}^2}$.

For the other two cases (Line 3 and Line 5), the chosen set can be identified in $O(n)$. Therefore the overall runtime of MEDIUMVALUEL$_1^+$ is $n^{O(1/p_{\min}^2)}$. $\qquad\square$

### B.8.3 Proof of Lemma 5

Recall from (1) that the objective is computed as $U([n]) = R([n]) - \lambda \cdot V([n])$, with

$$R([n]) := \sum_{i \in [n]} p_i x_i$$

$$V([n]) := \mathbb{E}\, \rho\left( \sum_{i \in [n]} Z_i, \, M \right).$$

It is clear that computing the reward term $R$ may be done in $O(n)$ operations. We now show that the penalty term $V$ can be computed in $O(n^2)$ operations.

We start by rewriting the term $V$ as:

$$V([n]) = \sum_{k=0}^{n} \mathbb{P}\left(\sum_{i \in [n]} Z_i = k\right) \cdot \rho(k, M) \tag{63}$$

For any integer $m \in \{0, 1, \ldots, n\}$, we define the $(m+1)$-dimensional vector $\{w^{(m)}\}_{k=0}^{m}$ by

$$w_k^{(m)} := \mathbb{P}\left(\sum_{i \in [m]} Z_i = k\right).$$

Since we assume that the relevant values of $\rho$ are known at the outset, it suffices to show that the probabilities involved in (63), or equivalently the $(n+1)$-dimensional vector $\{w_k^{(n)}\}_{k=0}^{n}$, can be computed in $O(n^2)$ operations.

We iteratively compute the vector of $\{w_k^{(m)}\}_{k=0}^{m}$ for $m \in \{0, 1, \ldots, n\}$. First, we observe that $w^{(0)} = 0$. Then we observe the iterative relation that for each $m \in [n]$ and $k \in \{0, \ldots, m\}$, we have

$$w_k^{(m)} = p_m \cdot w_{k-1}^{(m-1)} + (1 - p_m) \cdot w_k^{(m-1)}.$$

Hence, given the values of the $m$-dimensional vector $\{w_k^{(m-1)}\}_{k=0}^{m-1}$, computing each term $w_k^{(m)}$ takes $c$ operations, where $c$ is a universal constant. Hence, given the values of the $m$-dimensional vector $w^{(m-1)}$, it takes $c(m+1)$ operations to compute the $(m+1)$-dimensional vector $w^{(m)}$. Hence, the number of operations for computing the vector $w^{(n)}$, by iteratively taking $m \in \{1, 2, \ldots, n\}$, is

$$c \sum_{m=1}^{n} (m+1) = O(n^2),$$

completing the proof. $\qquad\square$

### B.8.4   Proof of Lemma 6

We re-index the items $\{p_i, x_i\}_{i \in [n]}$ in decreasing order of the value $x_i$, such that $x_1 \geq \ldots \geq x_n$.

First, note that if $p_1 > \beta \cdot \mu_S$, then $T = \emptyset$ satisfies the lemma. Clearly for this $T$ (33a) holds. Then because $R(S) \geq 0$ we also have

$$0 > \beta - \frac{p_i}{\mu_S} \geq \beta - \frac{1}{\mu_S} \geq \beta\left(1 - \frac{1}{\beta \cdot p_{\min}|S|}\right),$$

and so multiplying by $R(S)$ yields

$$R(T) = 0 > \beta\left(1 - \frac{1}{\beta \cdot p_{\min}|S|}\right) \cdot R(S),$$

satisfying (33b).

Otherwise we assume that $p_1 \leq \beta \cdot \mu_S$. We construct a set $T$ be selecting as many items as possible in the decreasing order of the value $x_i$, subject to the constraint that (33a) is satisfied. Formally, we consider the set $T := \{1, \ldots, t\}$, where

$$t := \max\left\{m \in [n] : \sum_{i=1}^{m} p_i \leq \beta \cdot \mu_S\right\}.$$

By the definition of $t$, the set $T$ satisfies (33a). It remains to show that the set $T$ also satisfies (33b).

If $\beta = 1$ then the resulting $T = S$ clearly suffices. Otherwise $\beta < 1$, and so we have $t < n$ (we assume that each item has strictly positive probability without loss of generality. By the definition of $t$, we have $\sum_{i=1}^{t+1} p_i > \beta \cdot \mu_S$. Equivalently,

$$\mu_T > \beta \cdot \mu_S - p_{t+1}. \tag{64}$$

In what follows, we use the following inequality that holds for any for $\{a_i\}_{i \in [n]}$ and $\{b_i\}_{i \in [n]}$ with $b_i \geq 0$:

$$\min_i \frac{a_i}{b_i} \leq \frac{\sum_i a_i}{\sum_i b_i} \leq \max_i \frac{a_i}{b_i}. \tag{65}$$

To see why this is true, note that for any $\{r_i\}_{i \in [n]}$ and $\{w_i\}_{i \in [n]}$, with $w_i \geq 0$ and $\sum_i w_i = 1$, we have

$$\min_i r_i \leq \sum_i w_i r_i \leq \max_i r_i.$$

We recover (65) by setting $r_i = \frac{a_i}{b_i}$ and $w_i = \frac{b_i}{\sum_i b_i}$.

Applying (65) yields

$$\frac{R(T)}{\mu_T} = \frac{\sum_{i \in T} p_i x_i}{\sum_{i \in T} p_i} \overset{(i)}{\geq} x_t \overset{(ii)}{\geq} \frac{\sum_{i \in S \setminus T} p_i x_i}{\sum_{i \in S \setminus T} p_i} = \frac{R(S \setminus T)}{\mu_S - \mu_T}, \tag{66}$$

where steps (i) and (ii) hold because the items are sorted in the decreasing order of $x_i$. Plugging $R(S) = R(T) + R(S \setminus T)$ into (66) and rearranging yields

$$R(T) \geq \frac{\mu_T}{\mu_S} R(S)$$
$$\overset{(i)}{\geq} \left( \beta - \frac{1}{\mu_S} \right) R(S)$$
$$\overset{(ii)}{\geq} \left( \beta - \frac{1}{p_{\min} \cdot |S|} \right) R(S),$$

where step (i) is true by (64), and step (ii) follows again from the fact that $\mu_S \geq p_{\min}|S|$. Hence, the set $T$ satisfies (33b), completing the proof. $\qquad \square$

### B.8.5  Proof of Lemma 7

To begin, we partition $S$ into "high," "bucketable," and "leftover" items according to their $p_i$ so that $S = H \sqcup B \sqcup L$ in Algorithm 7.

---

**Algorithm 7** PARTITION

---

**Require:** $S \in [n], p \in [0,1]^n$
**Ensure:** A partition $S = H \sqcup B \sqcup L$ with $B = D_1 \sqcup D_2 \sqcup D_3$
1: $H \leftarrow \{i \in S : p_i \geq \frac{1}{4}\}$
2: $L \leftarrow \{\}$
3: $D_1, D_2, D_3 \leftarrow \{\}$
4: **for** $\ell = 2, \ldots, \left\lceil \log_2\left(\frac{1}{p_{\min}}\right) \right\rceil - 1$ **do**
5: $\quad B^\ell \leftarrow \{i \in S : 2^{-(\ell+1)} \leq p_i < 2^{-\ell}\}$
6: $\quad$ **for** $j = 0, \ldots, \left\lfloor \frac{|B^\ell|}{3} \right\rfloor - 1$ **do**
7: $\quad\quad B_j^\ell \leftarrow \{b_{3j+1}^\ell, b_{3j+2}^\ell, b_{3j+3}^\ell\}$
8: $\quad\quad D_1 \leftarrow D_1 \cup \{b_{3j+1}^\ell\}$
9: $\quad\quad D_2 \leftarrow D_2 \cup \{b_{3j+2}^\ell\}$
10: $\quad\quad D_3 \leftarrow D_3 \cup \{b_{3j+3}^\ell\}$
11: $\quad L \leftarrow L \cup \{b_{3\lfloor\frac{|B^\ell|}{3}\rfloor+1}^\ell, \ldots, b_{|B^\ell|}^\ell\}$
12: **return** $S = H \sqcup B \sqcup L$ with $B = D_1 \sqcup D_2 \sqcup D_3$

---

Algorithm 7 first let $H = \{i \in S : p_i \geq \frac{1}{4}\}$, the high-probability items. Next consider the collection of buckets $B^\ell = \{i \in S \setminus H : 2^{-(\ell+1)} \leq p_i < 2^{-\ell}\}$. Note that the number of buckets is at most $\log_2\left(\frac{1}{p_{\min}}\right)$. Form the contents of each $B^\ell$ into groups of three, $\{B_j^\ell\}_j$ (that is, the set $\left|B_j^\ell\right| = 3$ for

each $j$. If the number of items in $B^\ell$ is not divisible by 3, we leave them to $L$). Let $B = \cup_\ell \cup_j B_j^\ell$, and let $L$ be the leftover $L := S \setminus (H \cup B)$ which do not belong to groups of three.

Next note that $R(H) + R(B) + R(L) = R(S)$ and that $V(H), V(B), V(L) \leq V(S)$. We handle the cases when each of these is large separately.

**Case 1:** $R(H) \geq \frac{R(S)}{3}$. If $|H| \leq \frac{24}{p_{\min}}$ then the set $H$ satisfies (35a). We now consider the case $|H| > \frac{24}{p_{\min}}$, and construct a set $T \subseteq H$ that satisfies (35b).

Applying Lemma 6 with $k = 6$ yields a set $T \subseteq H$ such that

$$\mu_T \leq \frac{1}{6}\mu_H \tag{67a}$$

and

$$R(T) \overset{(i)}{\geq} \frac{1}{6} \cdot \left(1 - \frac{6}{p_{\min} \cdot |H|}\right) \cdot R(H) \overset{(ii)}{\geq} \frac{1}{8}R(H), \tag{67b}$$

where step (i) follows from Lemma 6 and step (ii) is true by the assumption that $|H| > \frac{24}{p_{\min}}$. By the definition of $H$, we have $p_i \geq 1/4$ for each $i \in H$, and hence

$$|T| \leq 4\mu_T \overset{(i)}{\leq} \frac{2}{3}\mu_H \leq \frac{2}{3}\mu_S \overset{(ii)}{\leq} M,$$

where step (i) is true by (67a), and step (ii) is true by the assumption that $\mu_S \leq \frac{3}{2}M$. Hence, we have $V(T) = V(T') = 0$. By the rounding procedure, we have $R(T') = R(T)$. Therefore,

$$U(T') = U(T) = R(T) \overset{(i)}{\geq} \frac{1}{8}R(H) \overset{(ii)}{\geq} \frac{1}{24}R(S) \geq \frac{1}{24}U(S), \tag{68}$$

where step (i) is due to (67b) and step (ii) is true by the assumption of this case. Hence, the set $T$ satisfies the condition (35b).

**Case 2:** $R(L) \geq \frac{R(S)}{3}$. Recall that the number of buckets is at most $\log_2(\frac{1}{p_{\min}})$. Since there are at most two elements in $L$ from each bucket, the number of items in $L$ is at most $|L| \leq 2\log_2(\frac{1}{p_{\min}}) < \frac{2}{p_{\min}}$, satisfying condition (35a).

**Case 3:** $R(B) \geq \frac{R(S)}{3}$. Further partition $B$ into three equal-sized sets $B = D_1 \sqcup D_2 \sqcup D_3$ by arbitrarily assigning each member of each bucket-group $B_j^\ell$ to a distinct $D_\ell$, as performed in each iteration of line 6. Without loss of generality, assume that $D_1$ has the maximum reward among these three sets, namely $R(D_1) \geq \max\{R(D_2), R(D_3)\}$, so that $R(D_1) \geq \frac{R(B)}{3} \geq \frac{R(S)}{9}$.

In what follows, we first show that the set $D_1$ satisfies (35b) under the assumption

$$V(D_1') \leq V(S). \tag{69}$$

Then we show that assumption (69) always holds.

**Proving (35b) for set $D_1$.** For the reward term, we have

$$R(D_1') = R(D_1) \geq \frac{1}{9}R(S).$$

For the penalty term, recall that we assume $\lambda \cdot V(S) \leq \frac{1}{15}R(S)$. Combining the reward term and the penalty term, we have

$$\begin{aligned}
U(D_1') &= R(D_1') - \lambda \cdot V(D_1') \\
&\overset{(i)}{\geq} \frac{1}{9}R(S) - \lambda \cdot V(S) \\
&\geq \frac{1}{9}R(S) - \frac{1}{15}R(S) \\
&\geq \frac{1}{24}U(S),
\end{aligned}$$

where step (i) uses assumption (69). Hence, the set $D_1'$ satisfies condition (35b). It remains to prove assumption (69).

**Proving** (69). For any sets $S_1$ and $S_2$, we say that $S_1$ stochastically dominates $S_2$, if the random variable $\sum_{i \in S_1} Z_i$ stochastically dominates the random variable $\sum_{i \in S_2} Z_i$. Namely, for any $t \in \mathbb{R}$, we have

$$\mathbb{P}\Big( \sum_{i \in S_1} Z_i \geq t \Big) \geq \mathbb{P}\Big( \sum_{i \in S_2} Z_i \geq t \Big)$$

Since the one-sided loss $L_1^+$ is nondecreasing, it can be verified that if $S_1$ stochastically dominates $S_2$, then $V(S_1) \geq V(S_2)$.

By construction we have $B \subseteq S$, and hence $S$ stochastically dominates $B$. If $B$ stochastically dominates $D_1'$, then we have

$$V(D_1') \leq V(B) \leq V(S),$$

proving (69). It remains to prove that $B$ stochastically dominates $D_1'$.

For each bucket group $B_z = B_j^\ell$, let $B_z = \{p_1, p_2, p_3\}$ with $p_1 \in D_1$. Then let the associated random variables be

$$X_z := \mathrm{Ber}(q_1) \qquad \text{and} \qquad Y_z := \mathrm{Ber}(p_1) + \mathrm{Ber}(p_2) + \mathrm{Ber}(p_3),$$

where $q_i$ is obtained by rounding $p_i$ up to the nearest power of two. Note that $\sum_{i \in D_1'} Z_i' = \sum_z X_z$, where the $Z_i'$ are the realizations derived from the rounded probabilities, and $\sum_{i \in B} Z_i = \sum_z Y_z$. Moreover, $\{X_z\}_z$ are independent, and $\{Y_z\}_z$ are independent. It suffices to show the stochastic dominance of $Y_z$ over $X_z$ for each bucket group $b_z$, and then the stochastic dominance of $B$ over $D_1'$ follows.

To show the stochastic dominance of $Y_z$ over $X_z$, we consider the probabilities

$$\mathbb{P}(X_z = 0) = 1 - q_1$$
$$\mathbb{P}(Y_z = 0) = (1 - p_1)(1 - p_2)(1 - p_3),$$

and show that $\mathbb{P}(X_z = 0) \geq \mathbb{P}(Y_z = 0)$. By the construction of each bucket group $B_\ell$, we have $p_1, p_2, p_3 \in [2^{-(l+1)}, 2^{-l})$ and hence $q_1 = 2^{-l}$. Consequently, we have $p_1, p_2, p_3 \geq \frac{q_1}{2}$. We have

$$\mathbb{P}(X_z = 0) = 1 - q_1 \overset{(i)}{\geq} \Big( 1 - \frac{q_1}{2} \Big)^3 \geq (1 - p_1)(1 - p_2)(1 - p_3) = \mathbb{P}(Y_z = 0).$$

where it can be verified that step (i) holds for every $q_1 \in [0, \frac{1}{4}]$. Hence, $Y_z$ stochastically dominates $X_z$, completing the proof of (69). $\square$

### B.8.6 Proof of Lemma 8

For notational simplicity, we assume $\lambda = 1$ without loss of generality, and denote the random variable $T := \sum_{i \in S} Z_i$. We write the penalty term $V(S)$ as

$$V(S) = \mathbb{E}\, (T - M)_+$$
$$= \sum_{i=1}^{\infty} i \cdot \mathbb{P}\Big( T = M + i \Big), \tag{70}$$

We consider the probability that the value of $T$ lies in each interval $(M + ik, M + (i+1)k]$, for each integer $i \geq 0$. We have

$$V(S) \leq \sum_{i=0}^{\infty} (i+1)k \cdot \mathbb{P}\Big( M + ik < T \leq M + (i+1)k \Big)$$

$$\leq k \cdot \sum_{i=0}^{\infty} (i+1) \cdot \mathbb{P}\Big( T > M + ik \Big).$$

We bound each term $\mathbb{P}\left(T > M + ik\right)$ by Hoeffding's inequality. We have $\mathbb{E}[T] = \mu_S \leq M$ by assumption. Hence, by Hoeffding's inequality,

$$
\begin{aligned}
\mathbb{P}\left(T > M + ik\right) &\leq \operatorname{Exp}\left(-\frac{2(M + ik - \mu_S)^2}{|S|}\right) \\
&= \operatorname{Exp}\left(-\frac{2(M - \mu_S)^2}{|S|}\right) \cdot \operatorname{Exp}\left(-\frac{4(M - \mu_S)ik + 2(ik)^2}{|S|}\right) \\
&\leq \operatorname{Exp}\left(-\frac{2(M - \mu_S)^2}{|S|}\right) \cdot \operatorname{Exp}\left(-\frac{4(M - \mu_S)ik}{|S|}\right).
\end{aligned}
\tag{71}
$$

Plugging (71) into (70) yields

$$
\begin{aligned}
V(S) &\leq k \cdot \operatorname{Exp}\left(-\frac{2(M - \mu_S)^2}{|S|}\right) \cdot \sum_{i=0}^{\infty}(i+1) \cdot \operatorname{Exp}\left(-\frac{4(M - \mu_S)ik}{|S|}\right) \\
&\leq k \cdot \operatorname{Exp}\left(-\frac{2(M - \mu_S)^2}{|S|}\right) \cdot \sum_{i=0}^{\infty}(i+1) \cdot \left(e^{-\frac{4(M-\mu_S)k}{|S|}}\right)^i \\
&\overset{\text{(i)}}{=} k \cdot \operatorname{Exp}\left(-\frac{2(M - \mu_S)^2}{|S|}\right) \cdot \left(1 - e^{\frac{-4(M-\mu_S)k}{|S|}}\right)^{-2},
\end{aligned}
$$

where step (i) uses the fact that for any $0 < x < 1$, we have

$$
\sum_{i=0}^{\infty}(i+1)x^i = \sum_{t=0}^{\infty}\sum_{i=t}^{\infty}x^i = \sum_{i=0}^{\infty}\frac{x^t}{1-x} = \frac{1}{1-x}\sum_{i=0}^{\infty}x^t = \frac{1}{(1-x)^2}.
$$

$\square$

### B.9 Proof of Theorem 5

We fix any arbitrary problem instance $(x, p, \lambda, M)$ for the $L_1$ loss, and problem instance $(x', p, \lambda', M)$ for the $L_1^+$ loss, with

$$
\begin{aligned}
x'_i &:= x_i - \lambda \\
\lambda' &:= 2\lambda.
\end{aligned}
$$

We fix any arbitrary $S \subseteq [n]$, and demonstrate the desired equality

$$
U_{L_1}(S) = U_{L_1^+}(S') - \lambda \cdot M.
\tag{72}
$$

by induction on the number of elements in $S$. First, we consider $|S| = 0$, or equivalently $S = \emptyset$. Then it can be verified that

$$
\begin{aligned}
U_{L_1}(S) &= -\lambda M \\
U_{L_1^+}(S') &= 0,
\end{aligned}
$$

satisfying (72).

Next suppose that (72) holds for all set $S$ with $|S| \leq k$. We consider the marginal change to the objective when adding any item $j \notin S$ to the set $S$. Let $S_j =$ for some $S$ with $|S| < j$. The marginal change to the objective with the $L_1$ loss is

$$
\begin{aligned}
U_{L_1}(S \cup \{j\}) - U_{L_1}(S) &= p_j x_j + \lambda \cdot \mathbb{E}_{Z_{S \cup \{j\}}}\left[\left|\sum_{i \in S} Z_i + Z_j - M\right| - \left|\sum_{i \in S} Z_i - M\right|\right] \\
&= p_j x_j + \lambda \cdot p_j \cdot \mathbb{E}_{Z_S}\underbrace{\left[\left|\sum_{i \in S} Z_i + 1 - M\right| - \left|\sum_{i \in S} Z_i - M\right|\right]}_{T}
\end{aligned}
\tag{73}
$$

Note that the term $T$ satisfies

$$T = \begin{cases} 1 & \text{if } \sum_{i \in S} Z_i \geq M \\ -1 & \text{if } \sum_{i \in S} Z_i < M. \end{cases} \tag{74}$$

Using the fact (74) in (73), we have

$$U_{L_1}(S \cup \{j\}) - U_{L_1}(S) = p_j x_j + \lambda \cdot p_j \left[ \mathbb{P}\Big( \sum_{i \in S} Z_i \geq M \Big) - \mathbb{P}\Big( \sum_{i \in S} Z_i < M \Big) \right]$$

$$= p_j x_j + \lambda \cdot p_j \left[ 2 \cdot \mathbb{P}\Big( \sum_{i \in S} Z_i \geq M \Big) - 1 \right]$$

$$= p_j (x_j - \lambda) + 2\lambda p_j \cdot \mathbb{P}\Big( \sum_{i \in S} Z_i \geq M \Big)$$

$$= p_j x'_j + \lambda' p_j \cdot \mathbb{P}\Big( \sum_{i \in S} Z_i \geq M \Big). \tag{75}$$

Using a similar analysis, the marginal change to the objective with the $L_1^+$ loss is

$$U_{L_1^+}(S \cup \{j\}) - U_{L_1^+}(S) = p_j x'_j + \lambda' \cdot \mathbb{E}_{Z_{S \cup \{j\}}} \left[ \Big( \sum_{i \in S} Z_i + Z_j - M \Big)_+ - \Big( \sum_{i \in S} Z_i - M \Big)_+ \right]$$

$$= p_j x'_j + \lambda' p_j \cdot \mathbb{P}\Big( \sum_{i \in S} Z_i \geq M \Big). \tag{76}$$

Combining (75) and (76) demonstrates that the marginal change is equal for the $L_1$ and $L_1^+$ losses, under their respective instances. Therefore, applying the induction hypothesis that (72) holds for all $S$ with $|S| \leq k$ completes the induction step. $\qquad \square$