# OpenReview forum: "Recruitment Strategies That Take a Chance"
_NeurIPS.cc/2022/Conference — NeurIPS 2022 Accept_

### Official Review · Reviewer_vVfn · 2022-07-10

**Rating:** 6
**Confidence:** 4
**Soundness:** 2 fair
**Presentation:** 3 good
**Contribution:** 3 good

**Summary:**

This paper proposes strategies to find the optimal batch of offers to make in a recruitment scenario, where the goal is to maximize the quality of the recruited candidates but there is a soft constraint on the maximum number of candidates that can be recruited (and it is uncertain whether candidates will accept an offer or not). When the penalty function that characterizes the soft constraint is based on the mean squared error, the authors provide a polynomial-time strategy that can be made arbitrarily close to being optimal; when the penalty function is one-sided and linear, they provide a polynomial-time strategy that has a constant approximation factor.

**Questions:**

In all experiments presented, xGreedy seems to be equivalent to, if not better than, LowValueL1+. What is the reason behind this result? Is it because LowValueL1+ was run with $\tau=0$? If so, how sensitive is LowValueL1+ to the choice of $\tau$ and how would one tune $\tau$ for their specific problem instance? Is it because $p_{\text{min}}$ is not low enough to see the advantage of LowValueL1+ being a constant-factor approximation (as opposed to xGreedy whose performance depends on $p_{\text{min}}$)? If so, how does the performance difference between xGreedy and LowValueL1+ varies with respect to $p_{\text{min}}$? Since the performance xGreedy is lower bounded in terms of $p_{\text{min}}$, I assume it should be possible to make LowValueL1+ arbitrarily better than xGreedy. Is it possible to show this empirically? I believe more experiments are needed to answer these questions.

Also, I think evaluating the performance of the strategy proposed for $L_2$ with respect to the greedy strategies would have still been valuable. I understand that $L_2$ can be made arbitrarily close to being optimal but how would that compare to the greedy strategies? LowValueL1+ had different theoretical guarantees that made it more preferable over the greedy strategies, but the experiments revealed that xGreedy could perform better than a practical implementation of LowValueL1+ for most problem instances considered.

**Limitations:**

One limitation is that the strategies proposed in the paper are specifically designed for $L_2$ or $L_1^{+}$ penalty functions whereas a user might want to design their own penalty function according to the characteristics of their own problem instance based on historical data.

**Strengths And Weaknesses:**

The problem seems to be well motivated. I especially like the explanation of why the sequential and hard-constrained setting studied previously in the literature fails to capture the more practical complexities of recruitment (as opposed to single-batch and soft-constrained setting considered in this paper). However, most of the evidence for this seems to be anecdotal, maybe it could be supported with more references.

The problem is explored fully. In particular, a practical strategy is proposed for both a two-sided penalty function (where recruiting fewer candidates than targeted is penalized explicitly) as well as a one-sided penalty function (where recruiting fewer candidates is not penalized explicitly, but it would implicitly lower the total quality of the recruited candidates). However, it is not discussed when would one strategy/penalty function would be preferred to the other. While a one-sided penalty function seems natural to me, it would have been nice to have examples where under-recruiting would require explicit penalty (on top of the opportunity cost of not recruiting as many candidates as possible).

The experiments seem to be the weakest part of the paper; I have put my questions regarding the experiments below.

---

> ### Author Response · Authors · 2022-08-02
> **Response to review of Paper3632 by Reviewer vVfn**
>
> ### Questions
>
> > In all experiments presented, xGreedy seems to be equivalent to, if not better than, LowValueL1+. What is the reason behind this result? Is it because LowValueL1+ was run with $\tau=0$? If so, how sensitive is LowValueL1+ to the choice of $\tau$ and how would one tune $\tau$ for their specific problem instance? Is it because $p_{min}$ is not low enough to see the advantage of LowValueL1+ being a constant-factor approximation (as opposed to xGreedy whose performance depends on $p_{min}$)? If so, how does the performance difference between xGreedy and LowValueL1+ varies with respect to $p_{min}$? Since the performance xGreedy is lower bounded in terms of $p_{min}$, I assume it should be possible to make LowValueL1+ arbitrarily better than xGreedy. Is it possible to show this empirically? I believe more experiments are needed to answer these questions.
>
> The reviewer's point is well taken. In response, we have included a broader range of experiments in Section E of the revised appendix. Note that even in our original experiments, xGreedy does not always outperform LowValueL$_1^+$: in particular, it underperforms relative to both LowValueL$_1^+$ and xpGreedy in the settings with intermediate penalty regularizer and no correlation (see Figure 1, center) and with high penalty regularizer and negative correlation (Figure 2, right). As $\tau$ controls the number of small solutions which are checked by brute force, choosing $\tau > 0$ matters most for settings when the target M is small. If M is large and there are few candidates with high $x_i$ and $p_i$, then we should not expect the choice of $\tau$ to impact the quality of the solution. Finally, it is possible demonstrate instances where LowValueL$_1^+$ dramatically outperforms xGreedy, by constructing distributions which mimic the instance used to show that xGreedy is at most a p-min approximation (as described in the proof of Remark 5 in Appendix A.2).
>
> > Also, I think evaluating the performance of the strategy proposed for $L_2$ with respect to the greedy strategies would have still been valuable. I understand that $L_2$ can be made arbitrarily close to being optimal but how would that compare to the greedy strategies? LowValueL1+ had different theoretical guarantees that made it more preferable over the greedy strategies, but the experiments revealed that xGreedy could perform better than a practical implementation of LowValueL1+ for most problem instances considered.
>
>
> Our additive inapproximability results notwithstanding, evaluating the greedy algorithms on instances with two-sided $L_2$ error may well shed light on how they interact with the problem. In response to the reviewer's question, we now include experiments for this objective in Section E.2 of the revised appendix. The new experiments suggest that xGreedy performs quite poorly relative to xpGreedy for two-sided objectives, and we include a tentative explanation of why this might be the case. We paste it here for convenience:
>
> "*We observe that xpGreedy seems to dramatically outperform xGreedy when the loss is two-sided. We provide an informal explanation, using the two-sided $L_2$ loss as an example. Under this loss, a candidate $i$ contributes $x_i p_i$ to the reward term of the objective, while contributing $p_i(1 - p_i)$ to the variance of the realized size. When faced with two candidates of equal value $x_i$, we should therefore at the margin prefer the candidate with the higher probability, since this candidate contributes less to the variance per contribution to the reward. Note that for sufficiently large $\lambda$ two-sided loss functions encourage algorithms to choose solutions expected size very close $M$, meaning that the variance and the two-sided $L_2$ loss are nearly equal. Here xpGreedy prefers this higher-probability candidate, while xGreedy is indifferent, explaining the superior performance of xpGreedy.*"

---

### Official Review · Reviewer_xufP · 2022-07-10

**Rating:** 7
**Confidence:** 4
**Soundness:** 3 good
**Presentation:** 3 good
**Contribution:** 3 good

**Summary:**

The paper investigates the problem of batch hiring in an academic recruitment setting. While previous works consider a hard budget constraint on the number of acceptance, the authors argue that this assumption is inconsistent in practice and carries computational hardness. Instead, they view the budget as a soft constraint with a penalty for overshooting the target. Under these assumptions, the paper shows that greedy heuristics demonstrate some desirable properties. Finally, the authors provide an algorithm that approximates the optimal solution within a constant factor in polynomial time.

**Questions:**

- While the greedy heuristic is a natural benchmark, is there any other algorithm/benchmark that can be used for comparison in this setting?

- Although the result for L_2 loss is discussed, does the same result hold for the two-sided L_1 loss?

**Limitations:**

The authors have adequately addressed the limitations and potential negative societal impact in their paper.

**Strengths And Weaknesses:**

Strengths:

- The paper provides a novel algorithm for the case of one-sided L_1 loss between the realized outcome and the target number of hires.

- The paper is well-written and well-organized.

- The related work section is sufficient.

- The discussion of greedy heuristics performances is sufficient and provides good baseline intuition for the algorithm.

- The theoretical guarantee of algorithm ONESIDEDL1+ is significant and improves upon the greedy heuristics in experiments.

- The theoretical claims are supported by discussions that provide intuition to the proof.

- The numerical experiments section contains a sufficient discussion of the setting and explanations for choices of hyperparameters. The results of these experiments support the theoretical claims made.

Weaknesses:

- There are a few inconsistencies in the numerical experiments section. In the description of Figure 1, the paper stated "n=50". However, in line 304, at the end of the line, the paper stated "n=40". In line 337, "\lambda = 1", there is no plot for such value. Instead, in Figure 2, there are plots for $\lambda$ with values 1.5, 5, and 30.

---

> ### Author Response · Authors · 2022-08-02
> **Response to review of Paper3632 by Reviewer xufP**
>
>
> ### Weaknesses
>
> > There are a few inconsistencies in the numerical experiments section. In the description of Figure 1, the paper stated "n=50". However, in line 304, at the end of the line, the paper stated "n=40". In line 337, "\lambda = 1", there is no plot for such value. Instead, in Figure 2, there are plots for  with values 1.5, 5, and 30.
>
> We used $n=50$ in the experiments. We appreciate the proofreading, and have corrected these typos in the revised version of the paper.
>
> ### Questions
>
> > While the greedy heuristic is a natural benchmark, is there any other algorithm/benchmark that can be used for comparison in this setting?
>
> We believe the presented greedy heuristics are intuitive and natural, and for additional comparison, we now also benchmark against OPT for small instances (Figure 3 and Figure 4 in Section E.1 of the revised appendix). For completeness, here are two more heuristics we considered. (1) pGreedy: selecting the best prefix of candidates when ordered by decreasing probability; and (2) random: selecting the best prefix of candidates when ordered randomly. pGreedy fails when the highest-probability candidates have very low values, and random fails when only a small proportion of candidates have high values and probabilities. We opted not to include these two heuristics due to these practical weaknesses. If there are suggestions for other compelling baselines, we are open to including them.
>
> > Although the result for L_2 loss is discussed, does the same result hold for the two-sided L_1 loss?
>
> Unfortunately, it does not. The two-sided $L_2$ loss is handled by writing the problem as an unconstrained binary quadratic program, with a constant rank for the quadratic coefficient matrix. This approach fails for the two-sided $L_1$ loss, and we did not identify other means to provide an approximation to it.

---

### Official Review · Reviewer_VdAs · 2022-07-11

**Rating:** 8
**Confidence:** 3
**Soundness:** 4 excellent
**Presentation:** 4 excellent
**Contribution:** 3 good

**Summary:**

The paper considers a recruitment problem in which a agent can make offers to a set of possible candidates. Each candidate is characterized by a value and a probability of accepting the offer. The agent must make a single batch of offers, choosing a subset of the possible candidates. The objective is to maximize the expected value associated to the candidates that accept, minus a penalty term the depends on the deviation from a target number of desired acceptances.
They consider different types of deviations: $L_1$ and $L_2$ losses, and $L_1^+$ and $L_2^+$ losses in which recruiting less candidates that the target number is not penalized.
The authors characterized the computational complexity of computing optimal strategies and provide approximation results when the problem is intractable.
Finally, they perform an experimental analysis.

**Questions:**

No questions.

**Limitations:**

Yes

**Strengths And Weaknesses:**

The paper introduces a new computational problem. A similar problem was studied with sequential interaction, while this paper considers a different interaction with a single batch of  offers. This model is more realistic in many settings.

The paper is well written, easy to follow, and provides a detailed analysis of previous works.

The technical results are technically involved and provide a complete picture of the problem.

---

> ### Author Response · Authors · 2022-08-08
> **Response to Review of Paper3632 by Reviewer VdAs**
>
> We are grateful for the kind review.

---

### Official Review · Reviewer_6MKo · 2022-07-12

**Rating:** 5
**Confidence:** 3
**Soundness:** 3 good
**Presentation:** 2 fair
**Contribution:** 3 good

**Summary:**

The paper considers a problem of academic recruitment where the committee aims to select a set of candidates that maximizes the total expected reward minus the penalty caused by deviation from the target (i.e., over-hiring or under-hiring). It considers three simple greedy algorithms: (1) PGREEDY: selecting candidates in order based on their likelihood to accept the offer $p_i$; (2) XGREEDY: selecting candidates in order based on their value (qualification) $x_i$; (3) XPGREEDY: selecting candidates in order based on $x_i\cdot p_i$. The authors conduct the accuracy analysis of these greedy algorithms under certain settings and show both positive and negative results. For different penalty functions (i.e., one-sided and two-sided linear/square loss), the paper also proposes algorithms that attain fully polynomial-time approximation and constant-factor approximation to the optimal objective.

**Questions:**


1. In lines 215-218, the authors explained briefly why it’s difficult to analyze the algorithms under one-sided L2+ loss. Can you elaborate more? What does “quadratic factorization” mean and why it is an important property the objective needs to have? Why linearity is necessary for analyzing the one-sided loss?

2. In experiments, the authors construct synthetic data based on the approach of Purohit et al. Can you explain why this is good modeling? How will the algorithms perform on data with other distributions?

3. The experiments show that when $x$ and $p$ are negatively correlated, the performance of all algorithms drops significantly compared to positive and no correlation (Figure 1). Does it suggest that algorithms should be developed by incorporating such a negative correlation?

4. In practice, penalty functions may deviate from one-sided linear functions. While the performance of algorithms is theoretically guaranteed under a one-sided linear function, they can still be evaluated empirically on other types of loss functions. I suggest authors conduct experiments on settings with other loss functions to examine the robustness of the algorithms.

5. Algorithm 2, missing “{“ in lines 1-3; what are the notations $x|N_L$, $p|N_L$?

**Limitations:**

Yes

**Strengths And Weaknesses:**

Strengths:

1. The paper studies an interesting problem. In practical academic recruitments, multiple offers are typically made in parallel and target number of positions is typically treated as a soft constraint. This is captured in the model considered in this work.
2. The paper considers several simple greedy algorithms and conducts theoretical analysis. These results provide insights for better hiring practices.


Weaknesses:

1. My biggest concern is the restrictive conditions/assumptions of the results. The analytical results highly rely on the structure of penalty functions (i.e., induced cost as a function of the target and the actual number of hires). The paper primarily considers two types of penalty function (one-sided linear loss and two-sided square loss), and most results and algorithms are limited to these function types. In practice, we don’t know the exact penalty function, and we see from the results that different penalty functions may lead to completely different conclusions. As a result, the proposed method may not be easily deployed in practice.
2. The experiments only validate the proposed algorithms in synthetic settings; there is no real-world data that can be used for validation. Moreover, there is no other baseline method to compare with. The experiments show that when the candidate’s value $x_i$ and probability $p_i$ are negatively correlated (a realistic setting), the accuracy seems to be very low (e.g., in Figure 1, accuracy drops by almost half compared to the positive correlation case). It’s not clear how serious such a decrease is. What is the best attainable objective value $U(S*)$ for each target? It is important to highlight $U(S*)$ in the figures.

---

> ### Author Response · Authors · 2022-08-02
> **Response to Review of Paper3632 by Reviewer 6MKo**
>
> We believe we can provide satisfying answers to some of the reviewer's main questions and concerns.
>
> **Weakness 1**
>
> Our perspective on this issue is different. We agree that in practice we don't know the penalty function — but this actually aids the deployment of our methods. Instead of coming up with an exact penalty function (which we all agree is impossible), a user of our method would choose a "standard" penalty function whose structure seems to best match the scenario at hand. If the user doesn't know what penalty function to choose either, they may try different ones and inspect how their solutions differ. From this viewpoint, the goal should be to provide a rich menu of penalty functions to choose from; our results take significant steps in this direction.
>
> **Weakness 2 and Question 3**
>
> Finding the optimal solution appears to be intractable on many instances. However, in response to the reviewer's comment, we have included figures which compare our various algorithms with the optimal solution on smaller instances (where we can compute the optimal solution by brute force) in Section E.1 of the revised appendix. The figures demonstrate that, on these small instances, the algorithms are close to optimal.
>
> The drop in objective function for negative correlation does not point to a weakness of the algorithms. Rather, this is a fundamentally more difficult setting, and the optimum is indeed lower (as evidenced by the experiments in Section E.1 of the appendix). The reason is quite intuitive: In general, we desire high values and high probabilities (i.e., less variance). Under positive correlation, these two criteria coincide. The high-value candidates have high probabilities, so we recruit many high-value candidates whose probabilities are close to one, "filling up" the capacity while introducing very little variance for the penalty term. Conversely, under negative correlation, high value candidates have low probabilities, introducing a fundamental trade-off as for whom to select.
>
>
> **Question 1**
>
> By quadratic factorization we mean rewriting the two-sided $L_2$ objective (Equation 3) as the objective of a quadratic program (Equation 5 in Appendix B). That is, the two-sided $L_2$ objective is rewritten in the form $x^T A x + c^Tx + d$, where $x_i$ is the binary decision variable for whether to select candidate $i$. Obtaining this form with a matrix $A$ of constant rank is what allows us to construct an FPTAS for the two-sided $L_2$ objective. The $L_2^+$ objective does not take this form, and this approach fails.
>
> Note that we do not claim in Line 215-218 that the one-sided loss $L_2^+$ is necessarily inapproximable due to its nonlinearity; only that our algorithm for $L_1^+$ does not readily generalize. Concretely, our algorithm for one-sided $L_1^+$ attains a constant-factor approximation by first grouping  candidates based on their values and addressing each case separately (Algorithm 2). The low-value case (Algorithm 4 in Appendix C.1) relies on rounding the probabilities of the instance and bounding the effect of this rounding on the objective; the high-value case uses the fact that for the $L_1^+$ objective, if the value $x_i > \lambda$ then adding item $i$ to the selection always improves the objective. In either case, the same analysis does not apply to $L_2^+$.
>
> **Question 2**
>
> The distributions proposed by Purohit et al. capture our intuition about real-world phenomena. On the one hand, candidates may be high value because they are an excellent match with our institution, and are therefore more likely to accept an offer (positive correlation). On the other hand, candidates may be high value because they are objectively strong, and would therefore have many competing offers and be less likely to accept ours (negative correlation). Regarding performance on data from other distributions, it is difficult to give a precise answer without referring to a specific distribution. That said, we do note that our theoretical results — while more conservative — do not make specific distributional assumptions, and that is an advantage of our analysis.
>
> **Question 4**
>
> We agree that it is worth better understanding the empirical performance of our algorithms for other losses. In response to the reviewer's comment, we have added experiments evaluating the greedy heuristics against the one-sided $L_2$ loss function as well as two-sided loss funcitons to Section E.2 of the revised appendix. The experiments suggest that the performance of xGreedy relative to xpGreedy is similar under the one-sided $L_1^+$ and $L_2^+$ losses, with more marked differences between the two-sided $L_1$ and $L_2$ losses.
>
> **Question 5**
>
> The notation $x|_S$ is the restriciton of the $n$-dimensional vector $x$ to the subset $S\subseteq [n]$ of its coordinates. We thank the reviewer for pointing this out, and we now introduce this notation in the revised version of the paper (Line 250-251).

---

### Meta-Review · Area_Chair_MyJm · 2022-08-20

**Recommendation:** Accept
**Confidence:** Certain

**Metareview:**

Thank the authors for their submission.

The paper studies a hiring problem, arguably a more realistic formulation compared to prior work. An agent has access to a pool of candidates, each with its own value and probability of accepting a hiring offer. The goal is to select a batch of candidates as to maximize the cumulative value of the candidates accepting the offer minus a penalty term for deviating from a target number of candidates.

The problem is fully explored considering multiple types of penalty terms. Optimal algorithms are provided whenever possible and otherwise approximation algorithms are shown, all attained using simple greedy strategies. Synthetic experiments are also provided. The paper is well-written and easy to follow.

**Award:**

No

---

### Decision · Program_Chairs · 2022-09-14

Accept